

**Landslide displacement prediction using the GA-LSSVM model and time series analysis: a case study**
**of Three Gorges Reservoir, China**
Tao Wen[1], Huiming Tang[1,2]*, Yankun Wang[2], Chengyuan Lin[1], Chengren Xiong[2]
1 Faculty of Engineering, China University of Geosciences, Wuhan 430074, Hubei, People's Republic of China; 2 Three Gorges
Research Center for Geo-hazards of Ministry of Education, China University of Geosciences, Wuhan, Hubei 430074, People's
Republic of China
*Corresponding author: tanghm@cug.edu.cn
**Abstract** Predicting landslide displacement is challenging, but accurate predictions can prevent casualties and economic losses.
Many factors can affect the deformation of a landslide, including the geological conditions, rainfall, and reservoir water level.
Time series analysis was used to decompose the cumulative displacement of landslide into a trend component and a periodic
component. Then the least squares support vector machine (LSSVM) model and genetic algorithm (GA) were used to predict
landslide displacement, and we selected a representative landslide with step-like deformation as a case study. The trend
component displacement, which is associated with the geological conditions, was predicted using a polynomial function, and the
periodic component displacement which is associated with external environmental factors, was predicted using the GA-LSSVM
model. Furthermore, based on a comparison of the results of the GA-LSSVM model and those of other models, the GA-LSSVM
model was superior to other models in predicting landslide displacement, with the smallest root mean square error (*RMSE*), mean
absolute error (*MAE*), and mean absolute percentage error (*MAPE*). The results of the case study suggest that the model can
provide good consistency between measured displacement and predicted displacement, and periodic displacement exhibited good
agreement with trends in the major influencing factors.
**Keywords** landslide; displacement prediction; least squares support vector machine; genetic algorithm; reservoir water level;
rainfall

**1 Introduction**
In the Three Gorges Reservoir region, landslides are the main type of geo-hazard, and they cause critical harm to individuals
and property each year (Du et al. 2013; Yao et al. 2013; Lian et al. 2014; Cao et al. 2016). Therefore, geological surveying,
monitoring, landslide prevention and landslide prediction must be improved to mitigate the losses caused by landslides
(Kirschbaum et al. 2010; Miyagi et al. 2011; Ahmed 2013). A landslide can be regarded as a nonlinear and dynamic system that is
affected by external factors, such as geological conditions, rainfall, reservoir water levels, groundwater, etc. (Guzzetti et al. 2005;
Kawabata and Bandibas 2009). Due to the influences of external factors, deformation displacement of landslide generally exhibits
the same tendencies as the variations in external factors, which can result in misleading landslide prediction. The displacement
prediction of landslides is a major focus in the field of landslide research (Sassa et al. 2009; Du et al. 2013). Comprehensive
analyses of landslide response and displacement predictions of landslide based on external factors are effective methods that rely
on landslide deformation data. The accurate prediction of reservoir landslide processes is an important basis for early prevention,
and it can reduce the loss of property and lives (Corominas et al. 2005).
In recently years, grey system models, time series models, neural network models, extreme learning machines, support
vector machines (SVM), etc. have been widely used for landslide displacement prediction (Wang 2003; Pradhan et al. 2014;
Gelisli et al. 2015; Goetz et al. 2015; Kavzoglu et al. 2015). Previously, landslide susceptibility maps were assessed using a back
propagation artificial neural network and logistic regression analysis (Nefeslioglu et al. 2008). Additionally, dynamic time series
predictors were proposed based on echo state networks (Yao et al. 2013). Lian et al. (2013) used an extreme learning machine and
ensemble empirical mode decomposition to predict landslide displacement. Although these models were constructed based on
different algorithms, each has strengths and weaknesses. Grey system models are widely used in analyses of exponential time
series. However, for complex nonlinear slope displacement series, prediction results can yield considerable error (Yin and Yu 2007;
Sun et al. 2008). Additionally, autocorrelation coefficients, partial correlation coefficients and pattern recognition features are
difficult to determine via time series analysis (Brockwell and Davis 2013; Turner et al. 2015). The neural network method is a



powerful tool in landslide prediction (Liu et al. 2014; Lian et al. 2015). However, the conventional neural network has many
limitations, including overfitting and a shortage of theoretical guidance in the selection of the number of network nodes in the
hidden layer, which diminishes its prediction ability (Hwang et al. 2014). In addition, the neural network neglects practical issues
by using a pre-defined activation function. Compared with traditional learning algorithms, although extreme learning machines
are characterized by high generalization, good performance and fast computing speed, their output is different at different times
due to the use of randomly selected input (Lian et al. 2014). Thus, it is difficult to reflect large quantities of information
completely and predict landslide displacement accurately using these models because landslide displacement is actually a finite
time series.
The SVM model has strong generalization ability and can effectively overcome the limitations of other methods, including
small sample sizes, high dimensionality and nonlinearity. Many studies have illustrated the ability of SVM models to recognize
learning patterns, such as nonlinear regression, and obtain the global optimum solutions to these problems (Feng et al. 2004;
Marjanović et al. 2011; Micheletti et al. 2011; Hong et al. 2016). Although these problems can be transformed into quadratic
convex programming problems, the computation speed of the SVM model is slow when the training data set is large or the
dimensionality is high (Zhang et al. 2009). To overcome these inadequacies, we use the least squares support vector machine
(LSSVM) proposed by Suykens and Vandewalle (1999), which is a supervised learning model that has been widely applied in
other machine learning problems, such as function fitting. The LSSVM model uses the square sum of the least square linear
system error as the loss function and solves the problem by transforming it into a set of equations, which increases the solution
speed and reduces the required calculation resources (Suykens et al. 2002; Lv et al. 2013; Xu and Chen 2013; Zhang et al. 2013).
Additionally, this method yields good performance in pattern recognition and nonlinear function fitting. However, the selection of
parameters is crucial to developing an efficient LSSVM model due to its sensitivity to small variations in the parameters.
The genetic algorithm (GA) is a global optimization algorithm that uses highly parallel, random and adaptive searching
based on biological natural selection and optimization. Thus, the method is particularly suitable for solving complex and nonlinear
problems (Li et al. 2010; Ali et al. 2013). In this paper, the GA is selected as the method of parameter optimization in the LSSVM
due to its advantages in determining the unknown parameters that have great consistent between the predicted data and the
measured data. By introducing the GA, some key parameters of the LSSVM model can be ascertained automatically. Therefore,
we select the combination of the LSSVM model and the GA to predict landslide displacement.
Due to the influences of rainfall, reservoir water level and human activities on the monitoring data of landslide displacement,
most monitoring data series are incomplete or highly variable. These issues introduce uncertainty into the mathematical model and
increase the difficulty of prediction. To overcome this shortcoming and obtain the main error sources, a time series analysis of
displacement is conducted by decomposing the monitoring data series into several components (Du et al. 2013). Then, the
monitoring data series are simulated using the moving average method. This paper is organized as follows. The GA-LSSVM
model with time series analysis is described in Section 2. Taking a typical landslide with step-like deformation as an example,
details of the case study are introduced in Section 3. Model validation results based on landslide displacement prediction and the
prediction results of the GA-LSSVM model and other models are given in Section 4.
**2 Methodology**
**2.1 Time series analysis of displacement**
Cumulative displacement of landslide is caused by the joint effects of geological conditions (lithology, geological structure,
topography, etc.) and external environmental factors (rainfall, reservoir water level, etc.). The landslide displacement caused by
the former increases monotonically with time, which reflects the trend in cumulative displacement. However, the landslide
displacement induced by the latter is approximately periodic. Therefore, a landslide displacement sequence is an instability time
series with a periodic step-like characteristic. According to time series analysis, cumulative displacement can be decomposed into
three portions as follows:
$$y_t = p_t + q_t + \varepsilon_t \qquad (1)$$
where $y_t$ is the cumulative displacement, $p_t$ is the trend component displacement, $q_t$ is the periodic component displacement,



and  $\varepsilon_t$  is the random component displacement.
However, it is difficult to obtain relevant data regarding the random component (wind loads, car loads, etc.) due to the lack
of advanced monitoring methods. In this paper, the random component displacement is not considered. Therefore, we can simplify
the time series model as follows.

$$y_t = p_t + q_t \qquad (2)$$

The trend component can be extracted using the moving average method as follows:

$$A_i = \left\{ a_1, a_2, \cdots, a_j, \cdots, a_n \right\} \qquad (3)$$

$$\overline{p_t} = \frac{a_t + a_{t-1} + \cdots + a_{t-k-1}}{k} (t = k, k+1, \cdots, n) \qquad (4)$$

where  $A_i$  is the time series of cumulative displacement of the $i$th monitoring system ($i$=1, 2,…, $m$),  $a_j$  is the cumulative
displacement of the $i$th monitoring system at time $j$ ($j$=1, 2,…, $n$),  $\overline{p_t}$  is the extracted value of the trend component, and $k$ is the
moving average period.
The periodic component displacement can be acquired by subtracting the trend component displacement from the
cumulative displacement. Therefore, the time series model not only reflects the relationship between each component of
cumulative displacement but also provides mathematical and physical meaning for landslide displacement prediction.
**2.2 LSSVM**
The LSSVM model is a regression prediction method with nonlinear characteristics based on a statistical learning theory,
and it is regarded as an improved form of the SVM (Vapnik 1995; Abdi and Giveki 2013). First, after dividing the sample data
into training samples and testing samples, the training samples are plotted in a high-dimension feature space via nonlinear
mapping. Then, the optimal decision function model is obtained for the best-fitted results by training the sample data {$x_j$, $y_j$},
where $j$=1, 2, 3,…, $n$. The regression function of the LSSVM can be expressed as follows:
$$f(x) = W^T \varphi(x) + b \qquad (5)$$

where  $W^T$  is the weight vector,  $\varphi(x)$  is a nonlinear mapping function that maps the sample data into the feature space, $x$ is the
input, $y$ is the output, and $b$ is the offset.
By searching or a function $f(x)$ that adjusts the dispersion degree of the training samples, we can obtain a risk-minimized
solution. This solution can be written using the structural risk minimization principle:
$$\text{Minimize: } \frac{1}{2} W^T W + \frac{C}{2} \sum_{j=1}^n \xi_j^2 \qquad (6)$$

$$\text{Subject to: } y_j = W^T \varphi(x_j) + b + \xi_j (j = 1, 2, \cdots, n) \qquad (7)$$

where $C$ is a penalty factor representing the penalty degree of the training samples, $b$ is the offset, and  $\xi_j$  is the relaxation
factor.
Based on the Lagrange equation and duality theory, the optimization problem can be converted into a dual problem:
$$L(W, b, \xi, \alpha) = \frac{1}{2} W^T W + \frac{C}{2} \sum_{j=1}^n \xi_j^2 - \sum_{j=1}^n \alpha_j (W^T \varphi(x_j) + b + \xi_j - y_j) \qquad (8)$$





where $\alpha_j$ is the Lagrange multiplier.
The solution of the optimization equation is obtained by solving the partial differential form of the Lagrange equation with
respect to $W$, $b$, $\xi_j$, $\alpha_j$. The optimization equations are expressed as follows.

$$\begin{cases} \dfrac{\partial L}{\partial W} = 0 \Rightarrow W = \sum_{j=1}^{n} \alpha_j y_j \varphi(x_j) \\ \dfrac{\partial L}{\partial b} = 0 \Rightarrow \sum_{j=1}^{n} \alpha_j y_j = 0 \\ \dfrac{\partial L}{\partial \xi_j} = 0 \Rightarrow \alpha_j = C\xi_j \\ \dfrac{\partial L}{\partial \alpha_j} = 0 \Rightarrow y_j[W^T \varphi(x_j) + b] - 1 + \xi_j \end{cases} \quad (9)$$

The linear equations can be obtained by solving Eq. (9) with the elimination of $W$ and $\xi$:

$$\begin{bmatrix} 0 & I^T \\ I & ZZ^T + C^{-1}E \end{bmatrix} \begin{bmatrix} b \\ \alpha \end{bmatrix} = \begin{bmatrix} 0 \\ y \end{bmatrix} \quad (10)$$

where $y = [y_1, y_2, \cdots, y_l]^T$, $I = [1, \cdots 1]^T$, $\alpha = [\alpha_1, \alpha_2, \cdots, \alpha_l]^T$, $Z = [\varphi(x_1), \varphi(x_2), \cdots, \varphi(x_l)]^T$ and $E$ is the unit
matrix with $l$ dimensions.
Then, the regression prediction model of the LSSVM can be rewritten based on the above optimization problem:

$$f(x) = \sum_{j=1}^{n} \alpha_j K(x, x_j) + b \quad (11)$$

where $K(x_j, x)$ is a kernel function.
In the paper, we select the radial basis kernel function as the kernel function in the LSSVM model to obtain the optimal
solutions due to its strong nonlinear mapping ability and wide convergence domain (Min and Lee 2005; Altınel et al. 2015; Elbisy
2015; Farzan et al. 2015):

$$K(x_j, x) = \exp(-(x - x_j)^2)/(2\sigma) \quad (12)$$

where $\sigma$ is a parameter of the kernel function.
The parameter of the model $C$ and the parameter of the kernel function $\sigma$ significantly influence the prediction
performance. The parameter $C$ represents the error tolerance. The more accurate the parameter is, the higher the prediction
performance is, but this can lead to overtraining. The parameter $\sigma$ implicitly determines the spatial distribution of data mapping
in the new feature space. Therefore, some measures should be taken to optimize the LSSVM parameters.
**2.3 GA**
Currently, several intelligent algorithms are used to solve optimization problem, such as the GA (Li et al. 2010; Ali et al.
2013), grid algorithm (Lin 2001), particle swarm optimization (Vandenbergh and Engelbercht 2006) and genetic programming
(Garg and Tai 2011; Shen et al. 2012). However, compared with the GA, the grid algorithm is tedious and cannot yield
satisfactory results (Gu et al. 2011). For discrete optimization problems, particle swarm optimization performs poorly and often
yields local optima (Fei et al. 2009). In addition, genetic programming, which was developed by Koza (1992), provides solutions
to complex problems using evolutionary algorithms, and the method is typically expressed as a tree structure that consists of
terminals and functions; however, it is difficult to generate new individuals, which seriously affects the convergence rate (Garg et





al. 2014). In this paper, we select the GA to determine the best parameters ($C$ and $\sigma$) of the LSSVM for predicting landslide
displacement.
The GA is a computational model that simulates natural selection and the biological evolution processes of genetic
mechanisms. The GA provides solutions for complicated problems using evolutionary algorithms (Levasseur et al. 2008; Hejazi et
al. 2013). The typical genetic operations include selection, crossover and mutation.
Based on certain methods and theories, selection operations, such as the fitness-ratio selection algorithm, ranking algorithm,
Monte Carlo selection and tournament selection, are commonly used to choose a parental generation from a population based on
an individual's fitness value. Crossover operation can generate two new offspring by selecting random codes from two parents
and then exchanging their respective branches. Point mutation is commonly used as the mutation operator. By selecting a random
node from a parent, a new individual is generated by substituting the selected random node into another parent branch. A typical
genetic algorithm is shown in Fig. 1. Selection operations, crossover operations and mutation operations are probabilistic, and
with a probability of over 90%, crossover operations are the most widely used.
By randomly choosing selection, crossover and mutation operations and beginning with any initial population, a set of new
individuals with better fitness can be produced. Therefore, the GA will terminate when the fitness of an optimal individual reaches
the threshold value, the fitness of an optimal individual and the fitness of the population no longer increase, or the number of
iterations reaches the default threshold.

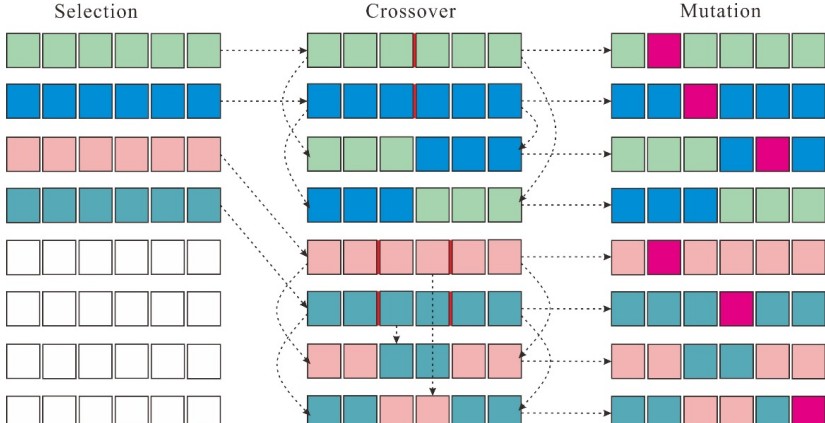

Fig. 1 Diagram of genetic operations
**2.4 GA-LSSVM model**
To obtain the best model, the parameters of the model must be carefully selected in advance (Duan et al. 2003). According
to some research results (Lessmann et al. 2005; Pourbasheer et al. 2009), the GA has the advantages of reducing the blindness of
artificial selection and enhancing the discrimination ability of the LSSVM model. Modeling with this method can achieve high
precision if the training samples are reliable. The sampling data used for landslide displacement prediction are continuous and
mutually dependent landslide data applicable for a specific method; thus, the data are essentially independent sampling data.
In this paper, the periodic component displacement is predicted by the GA-LSSVM model, which has higher accuracy than other
models due to the consideration of the trigger factors. MATLAB software is used to execute the model. The flowchart of the
GA-LSSVM model is presented in Fig. 2.





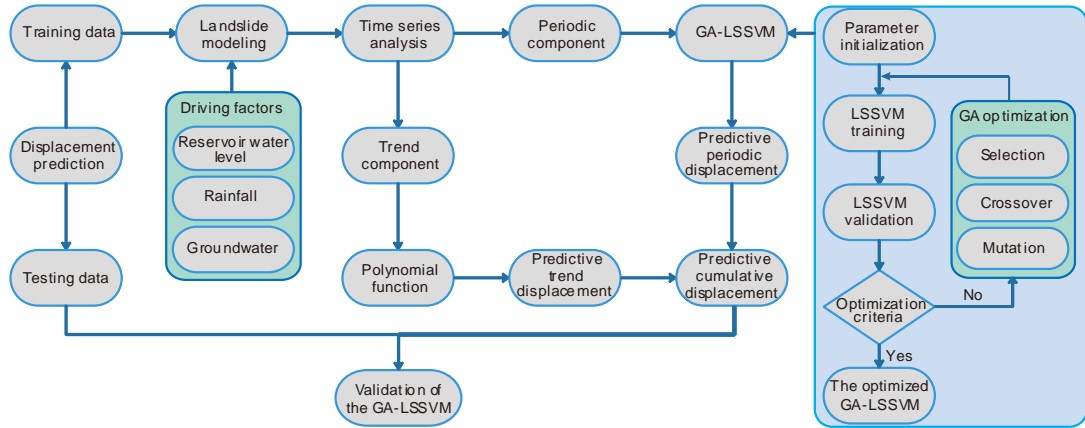

Fig. 2 The basic flowchart of the GA-LSSVM model, including the establishment of the GA-LSSVM model and the validation of
the model
**3 Case study: Shuping landslide**
**3.1 Geological conditions**
The Shuping landslide is located in Shazhenxi town, Zigui country, Hubei province, China, near the Yangtze River and
approximately 47 km into the upper reach of Three Gorges Dam (Fig. 3). The sliding direction of Shuping landslide is N11°E, and
the landslide presents a sector in a topographic map (Fig. 3). The reservoir water level in Fig. 3 is 166 m. The topography is
relatively flat, with a mean slope angle of 22°. The highest elevation of the landslide is 400 m above sea level. The head scarp of
the landslide reaches to the riverbed of the Yangtze River at 60 m in elevation. The landslide covers an area of approximately $54\times$
$10^4$ m$^2$, with an average length of 800 m in the longitudinal direction and an average length of 670 m in the transverse direction.
The landslide volume is $2070\times10^4$ m$^3$, with an average sliding surface depth of 40 m (Fig. 4). Fig. 4 shows the eight GPS
monitoring stations installed on the ground surface of the landslide, as well as four inclinometer monitoring holes. The bedrock is
mainly sandy mudstone. The strata comprise the Triassic Badong formation. The dip direction of the bedrock is between 120° and
165°, and the dip angle is between 10° and 35°. The landslide is divided into an eastern portion and a western portion, and the
materials of the landslide mainly include Quaternary deposits and soils containing silty clay and rock fragments with a loose and
disorderly structure (Fig. 5). Fig. 5 shows a longitudinal section of the eastern portion. The sliding surface is steep in the upper
area, which is located between the deposits and the bedrock.
Underground moisture beneath the landslide is primarily pore water flowing through a loose medium that consists of
colluviums, deposits, etc. After the water storage began in Three Gorges Reservoir in June 2003, the landslide deformation
became more active. Various external factors affect the landslide displacement, including rainfall, the reservoir water level, surface
water infiltration, groundwater, etc.
**3.2 Monitoring data and deformation characteristics of the landslide**
Field investigations revealed that there was no obvious deformation of this landslide before the first impoundment of the
reservoir on June 15, 2003. However, macroscopic cracks occurred in the landslide, including through roads and houses, after the
first impoundment. To measure the deformation characteristics and stability of the landslide, monitoring stations were built to
observe the interactions between different portions of the landslide. The monitoring methods include geodetic surveys, drilling,
meteorological observations and geological investigations. Thus, the development processes and evolution of the landslide can be
analyzed qualitatively using monitoring data from eight monitoring stations and four inclinometer monitoring holes located along
the longitudinal direction of the landslide (ZG85 to ZG90, SP-2 and SP-6, QZK1 to QZK4 in Fig. 4).



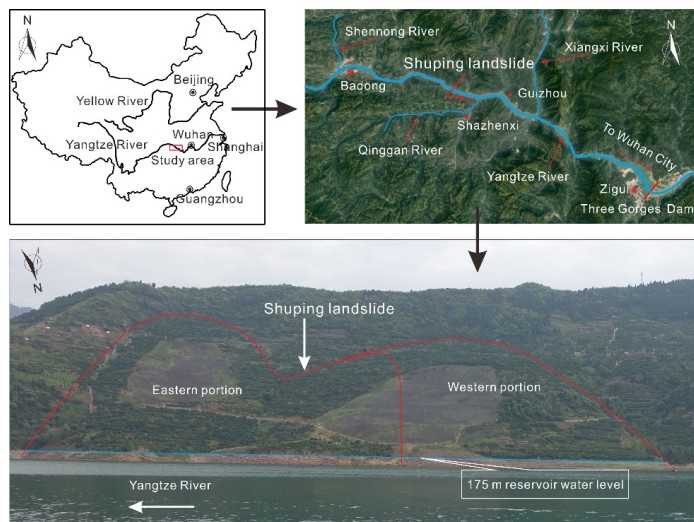

2    Fig. 3 Location of the study area and panorama of the Shuping landslide and landslide subzones

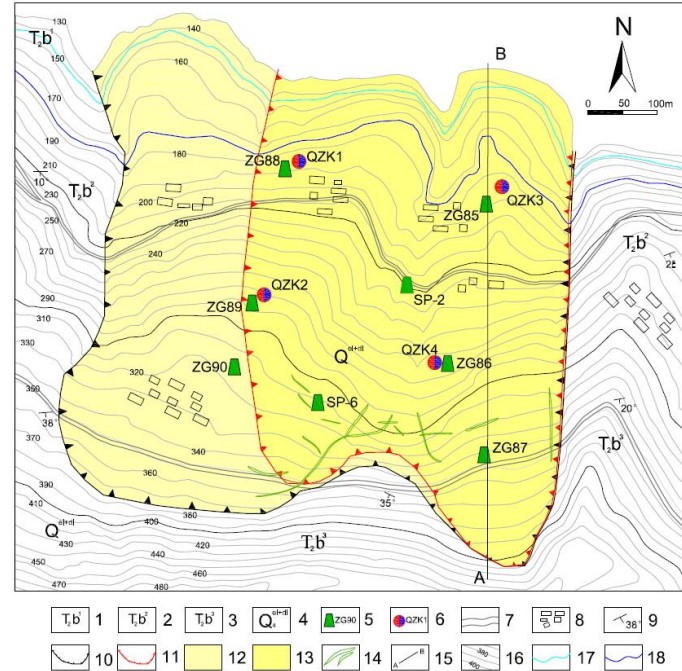

4    Fig. 4 Geology and deformation monitoring map of Shuping landslide





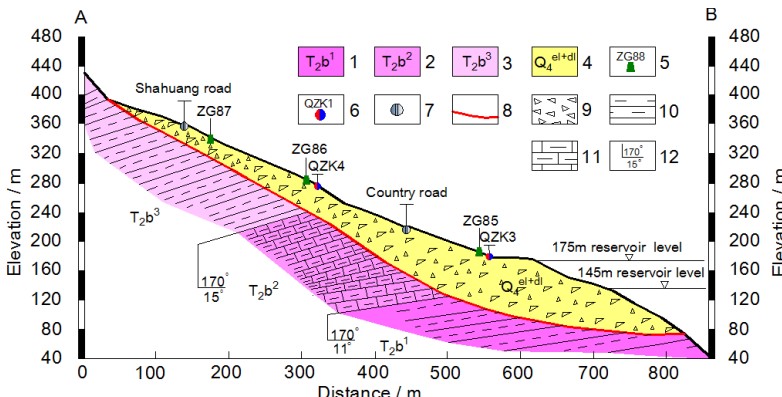

Fig. 5 Geological longitudinal section (line A-B in Fig. 4) of Shuping landslide

4        Fig. 6 shows the monitoring results between July 2003 and October 2013, including rainfall and reservoir water level, which

exhibit near step-like characteristics after the first impoundment. The displacements in the middle (ZG86) and head scarp (ZG85)
areas were greater than that in the back scarp (ZG87) area of longitudinal section A-B, and the displacements in the head scarp
(ZG88) and middle (ZG89) areas were greater than that in the back scarp (ZG90) area in the western zone. These observations
suggest that the displacement of landslide increased steadily, and Shuping landslide displayed retrograde style deformation from
the lower part to the upper part. The cumulative displacements at the monitoring stations located in the frontal areas were
relatively low, with an average value of 880 mm, and the cumulative displacements at the monitoring stations located in the
middle-rear areas were very high, with an average value of 3890 mm. Overall, landslide deformation in the eastern zone was
greater than that in the western zone. Based on the reservoir water level data and the displacements measured at eight monitoring
stations, the cumulative displacement rate increased after the initial impoundment. Due to the increased rainfall and decreased
reservoir water level between April and August each year, the cumulative displacement rises rapidly. Notable leap characteristics
can be observed in 2007, 2009, 2011 and 2012. The variations in reservoir water level and heavy rainfall reduced the matric
suction and the shear strength of soils and rocks. In addition, the uplift pressure, hydrostatic pressure and hydrodynamic pressure
acting on the landslide changed periodically. As a result, landslide deformation increased and the stability of the landslide
decreased.



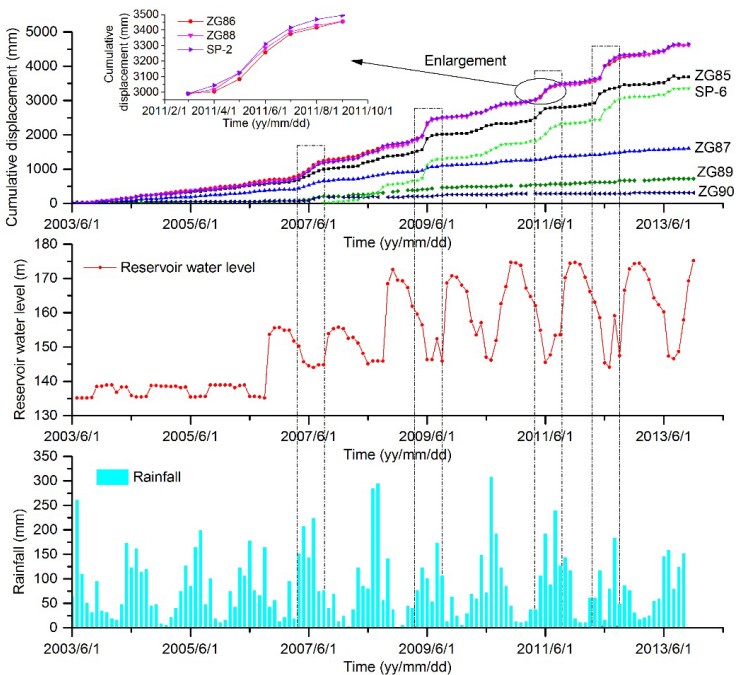

Fig. 6 The relationships between rainfall, reservoir water level and displacement

3        Many deformation or failure phenomena were observed in the Shuping landslide. These phenomena increase the risk of

endangering the lives of local inhabitants, as well as local property and infrastructure. In June 2003, a crack was generated in the
middle part of the landslide on the outside of a local road, as shown in Fig. 7(a). In 2006, the reservoir water level increased to
156 m for the first time. Fig. 7(b) shows that the crack gradually extended to a width of 10 cm within 3 months after finishing the
road in April 2007. In August 2008, after a heavy storm occurred, deformation and tension cracks developed in the eastern portion
of the landslide and affected houses, as shown in Fig. 7(c). Since 2009, the reservoir water level has increased gradually to 175 m.
In June 2009, the western portion of the landslide started cracking, with a maximum crack width of 20 cm and depth of 20-50 cm.
In addition, several tension cracks formed at the eastern landslide boundary. The tension cracks in the eastern portion are shown in
Fig. 7(d). In recent years, the cumulative deformation rate has remained low due to the relatively stable reservoir water level,
which has fluctuated between 145 m and 175 m.

13        Therefore, the macroscopic deformation characteristics suggest that deformation in the western portion of the landslide is

smaller than that in the eastern portion, and the Shuping landslide is affected by reservoir water level fluctuations and rainfall.
When rainfall increases abruptly and the reservoir water level drops between April and August annually, the landslide becomes
active, which increases landslide deformation. In other conditions, the landslide undergoes slow deformation at a constant speed.

17        In addition, groundwater, which is regarded as an active geologic agent, is one of the main factors that induces landslide

instability. In the rising phase of reservoir water level, the groundwater level gradually increases, with a slight lag behind the
increase in the reservoir water level. Conversely, the groundwater level decreases in the declining phase of the reservoir water
level. Moreover, the uplift pressure and seepage force of groundwater are dynamic processes that affect landslide stability.
Therefore, groundwater influences cumulative displacement.

22        Overall, the reservoir water level, rainfall and groundwater are the major factors that influence the cumulative displacement

of the Shuping landslide. The landslide displacement obviously increases when the reservoir water level decreases or when
rainfall is heavy and continuous because the materials in the sliding mass are degraded by the excess moisture and the additional
hydrodynamic pressure.



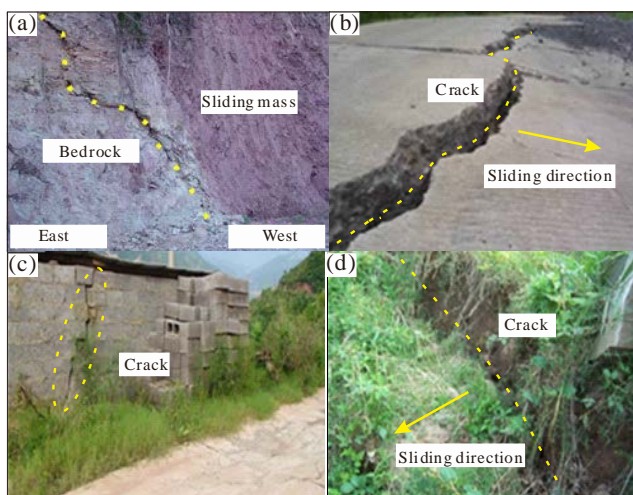

Fig. 7 Photographs of the ground cracks in the landslide (Ren et al., 2015): (a) crack in the middle of the landslide on the outside
of the local road, (b) failure state of the local road, (c) wall cracking and subsidence in the eastern portion, and (d) the tension
cracks in the eastern portion

6       Several years of monitoring data show that the landslide deformation differences are manifested in the ground surface, and

they display vertically distributed characteristics with elevation. In conclusion, the surface displacements below 200 m in
elevation are larger than those above 200 m, and deformation is largest close to 175 m, which is the upper limit of the reservoir
water level. This observation is due to the considerable influence of fluctuations in the reservoir water level on the landslide area
below 200 m. The deep deformation of the landslide exhibited distinct differences at different depths, as shown in Fig. 8.
Inclinometer monitoring holes QZK3 and QZK4, QZK1 and QZK2, which are located in the western portion of the landslide,
exhibited small deformation and similar deformation trends. Thus, their lateral displacement curves are not presented, and only the
curves of QZK3 and QZK4 are illustrated in this paper. The figures show that the sliding zones of QZK3 and QZK4 are located at
elevations of 70 m and 30 m, respectively. Furthermore, the displacement change in the shallow sliding zones of both QZK3 and
QZK4 is larger than that in the deep sliding zone.

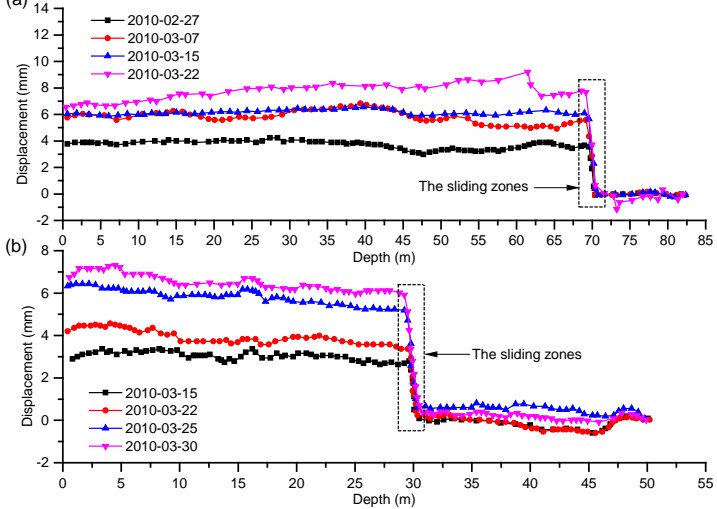

Fig. 8 Lateral displacements of Shuping landslide: (a) inclinometer monitoring hole QZK3 and (b) inclinometer monitoring hole



QZK4
**4 Landslide displacement prediction**
Based on the analysis of the deformation characteristics of Shuping landslide and the GA-LSSVM model above and due to
the obvious nonlinear and step-like deformation characteristics of monitoring stations ZG85, ZG86 and ZG87, we select only
these stations along longitudinal section A-B to verify and establish the prediction model. The model includes information
regarding rainfall, the reservoir water level, human activities and the long-term behavior of Shuping landslide. Because the
integrity of the data collected at monitoring points has an effect on the displacement prediction, the monitoring data from July
2003 to October 2013 are selected to explore landslide deformation. The data before October 2012 are used to train the
GA-LSSVM model, and the data after October 2012 are used to test the model.
**4.1 Prediction of the trend component displacement**
Due to the scheduling period of the reservoir and the rainfall cycle, we choose 12 months as the moving average period.
Because the curves of the trend component displacement versus time have quasi-linear and incremental characteristics, we use
polynomial functions to fit these curves and provide the best-fitted results. The predicted and measured results of the trend
component displacement at monitoring stations ZG85, ZG86 and ZG87 are shown in Figs. 9(a), 9(b) and 9(c), respectively. They
indicate that the polynomial function provides good prediction performance for the trend component displacement and the fitted
functions are expressed in Eqs. (13), (14) and (15).
$$p_{\mathrm{t}} = -0.0015t^3 + 0.4744t^2 - 8.4975t + 128.83 \quad R^2=0.9980 \qquad (13)$$
$$p_{\mathrm{t}} = -0.002t^3 + 0.604t^2 - 10.468t + 143.35 \quad R^2=0.9978 \qquad (14)$$
$$p_{\mathrm{t}} = -0.0015t^3 + 0.3088t^2 - 2.7227t + 29.832 \quad R^2=0.9976 \qquad (15)$$
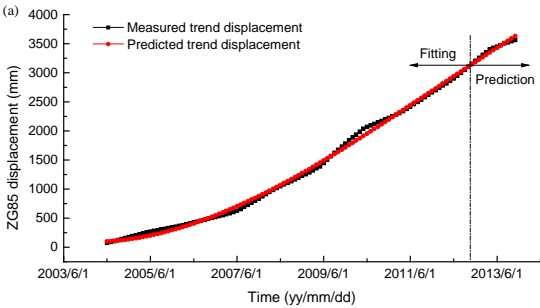 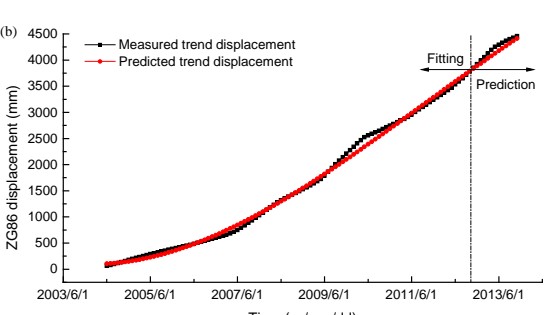
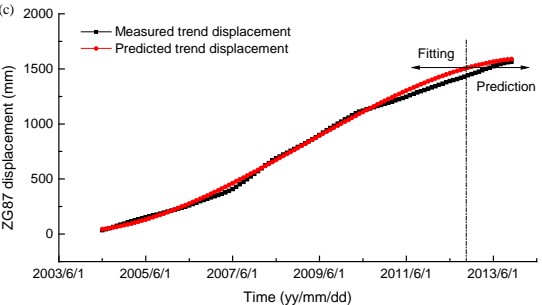
Fig. 9 Measured and predicted trend component displacement of Shuping landslide
**4.2 The predicted periodic component displacement**
The periodic component displacement is determined by subtracting the extracted trend component displacement from the
cumulative displacement. The periodic displacement and the major influencing factors are illustrated in Figs. 10 and 11. The





variations in the periodic displacement are consistent with those in the influencing factors. The reservoir water level, rainfall and
groundwater considerably influence the periodic displacement. For example, large periodic displacement can be observed in July
2009 and September 2012 when the landslide was affected by heavy rainfall and large variations in reservoir water level.
Although the variation in reservoir water level was small before April 2007, the periodic displacement still exhibited small
fluctuations due to the effects of rainfall and groundwater. After April 2007, several obvious peaks can be observed in the periodic
displacement-time curves during periods of decreasing reservoir water level. For example, the periodic displacement increased
from May to July 2009 and from May to September 2012. However, when the reservoir water level increased from 145 m to 175
m, the periodic displacement gradually decreased. The main reason for the above conditions was that the increase in the reservoir
water level increased the pressure on the surface of the landslide, thereby increasing the resistance force. Conversely, the sliding
force increased when the reservoir water level decreased. The periodicity of the rainfall also affected the periodic displacement.
The periodic displacement increased with increasing rainfall and reached a peak value in summer, which reflects a certain lag.
Groundwater depth was measured at the head scarp of the landslide at an elevation of 181 m in inclinometer monitoring hole
QZK3. The change in groundwater depth exhibits considerable agreement with rainfall and reservoir water level fluctuations, with
a slight lag observed for the latter. Due to the slight lag with the reservoir water level, groundwater increased the hydrodynamic
pressure during periods when the reservoir water level decreased or remained stable, which resulted in continuous deformation of
the landslide. Therefore, in the shallow groundwater zone, the periodic displacements measured at the three monitoring stations
exhibited considerable fluctuations. In conclusion, the results in Figs. 10 and 11 show that the reservoir water level is the factor
that most influences the periodic displacement.

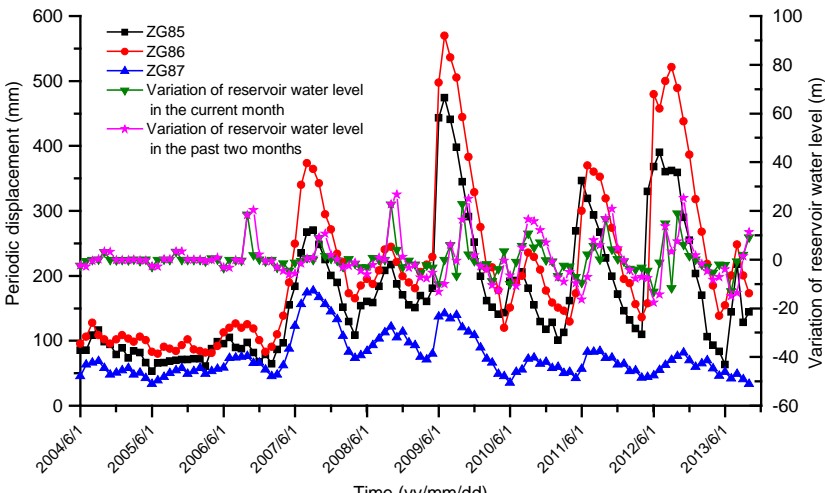

Fig. 10 The relationship between reservoir water level and the periodic displacement at GPS monitoring stations





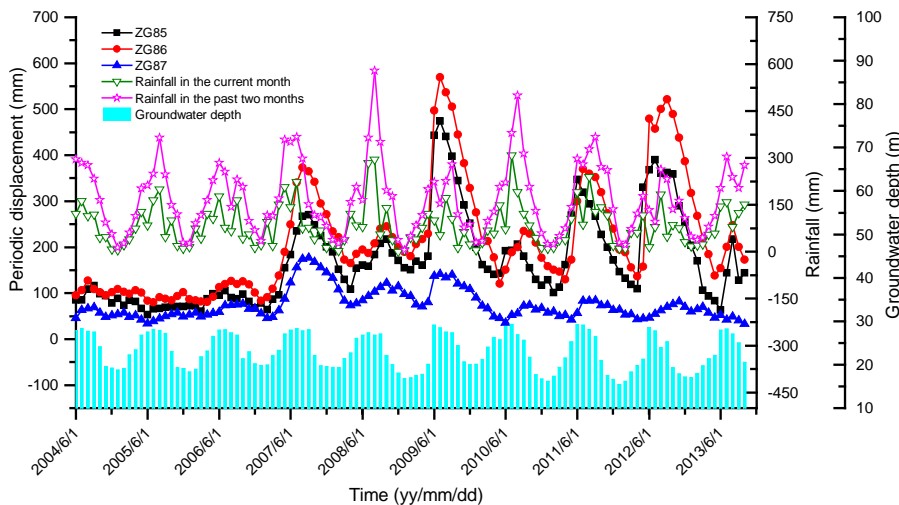

Fig. 11 The relationships between rainfall, groundwater depth and periodic displacement at GPS monitoring stations

3         The grey relational grade can represent the proximity degree between two series. If the trends in the two series are

consistent or the degree of synchronous change is high, then the relational grade associated with system development is large.
Otherwise, the relational grade is small. To remove the influence of dimensional data, data series must be normalized before
calculating the relational grades, including the series of periodic displacement, rainfall and reservoir water level changes. The
normalized formula can be expressed by Eq. (16):

$$\overline{y} = \frac{y - y_{\min}}{y_{\max} - y_{\min}} \qquad (16)$$

where $\overline{y}$ is the normalized value, $y$ is the original value, $y_{\max}$ is the maximum value of the data series, and $y_{\min}$ is the
minimum value of the data series.

11         Based on the grey relational analysis method, distinguishing coefficient is 0.5, and the relational grades between the

influencing factors and the periodic displacements are shown in Table 1. We can use the large grey relational grades as the input
variables in the GA-LSSVM model. When the relational grade is larger than 0.6, the influencing factor is closely correlated with
the periodic displacement, which suggests that the selection of the influencing factor for predicting periodic displacement is
reasonable (Wang 2003; Wang et al. 2004). Therefore, considering the characteristics of the periodic displacement and the
relational grades between variables, the cumulative rainfall in the current month, the cumulative rainfall in the past two months,
the reservoir water level, the variation in the reservoir water level in the current month, the variation in the reservoir water level in
the past two months, and groundwater depth are selected as input variables. Moreover, the periodic component displacement is
established as the output variable for use in the GA-LSSVM model.
Table 1 Relational grades between input variables and the periodic displacements

| Monitoring station | Relational grade | | | | | |
|---|---|---|---|---|---|---|
| | The cumulative rainfall in the current month | The cumulative rainfall in the past two months | The reservoir water level | The variation of The reservoir water level in the current month | The variation of The reservoir water level in the past two months | Groundwater depth |
| ZG85 | 0.700 | 0.705 | 0.763 | 0.797 | 0.768 | 0.718 |
| ZG86 | 0.682 | 0.691 | 0.756 | 0.794 | 0.770 | 0.714 |
| ZG87 | 0.692 | 0.705 | 0.724 | 0.794 | 0.780 | 0.720 |

21         The parameters of the LSSVM are optimized by the GA, including the best values of $C$ and $\sigma$. Table 2 shows the optimal



parameters of the LSSVM. The maximum generation threshold of the GA is 200, and the population number is 20. To validate the
prediction ability of the GA-LSSVM model, we compare the results of generalized regression neural network (GRNN) and back
propagation (BP) with two hidden layers with the result of the GA-LSSVM model. In this paper, the smoothing factor of the
GRNN is 0.48, and there are 10 nodes in one of the hidden layers and 11 nodes in the other hidden layer of the BP.
Table 2 Optimal parameters of the LSSVM model

| Number | Monitoring station | $C$ | $\sigma$ |
|---|---|---|---|
| 1 | ZG85 | 11.8234 | 6.4122 |
| 2 | ZG86 | 4.7346 | 8.0545 |
| 3 | ZG87 | 39.7819 | 5.7981 |

The prediction results of the periodic component displacement are shown in Fig. 12. The predicted values of the three
prediction models and the measured values are consistent and illustrate similar trends. However, the predicted values obtained
using the GA-LSSVM exhibit better agreement with observations than do those of the other methods. Notably, the advantages of
the model are clear from April 2013 to October 2013, as the periodic component displacement exhibited good agreement with the
major influencing factors during a period of heavy rainfall and large fluctuations in the reservoir water level.

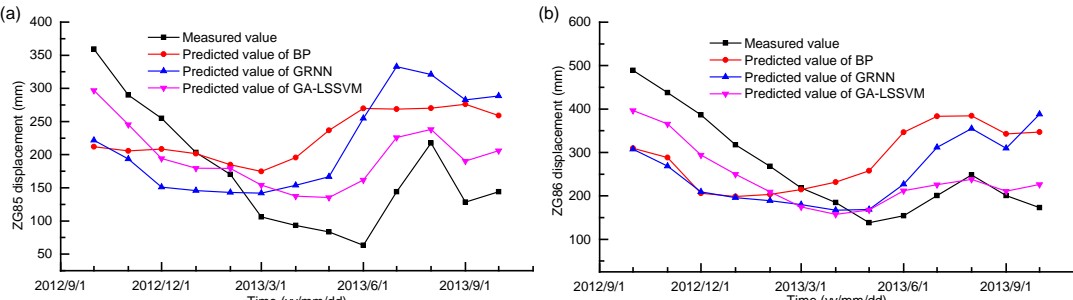

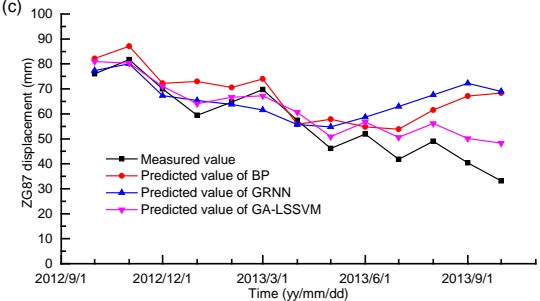

Fig. 12 Measured displacement and predicted periodic displacement of Shuping landslide
**4.3 Predicted cumulative displacement**
The predicted cumulative displacement is determined from the sum of the predicted trend displacement and the predicted
periodic displacement. The predicted cumulative displacements and the measured values are presented in Table 3, Table 4 and
Table 5 for monitoring station ZG85, ZG86 and ZG87, respectively. The results given in Table 3, Table 4 and Table 5 suggest that
the GA-LSSVM model has better prediction performance than the GRNN model and the BP model, with a smaller relative error.
Comparisons between the predicted values of cumulative displacement and measured values are shown in Fig. 13. The diagonal
line shows the best prediction result in Fig. 13. The results are underestimated if the predicted values are located below the
diagonal line, whereas the predicted values located above the line are overestimated. The predicted values from all the monitoring
stations show good consistency with the measured values, as shown in Fig. 13.
Table 3 Comparison between the predicted values of cumulative displacement and measured values at monitoring station ZG85

| Time | Measured | GA-LSSVM | GRNN | BP |
|---|---|---|---|---|



| | value (mm) | Predicted value (mm) | Relative error (%) | Predicted value (mm) | Relative error (%) | Predicted value (mm) | Relative error (%) |
|---|---|---|---|---|---|---|---|
| 2012/10/1 | 3460.208 | 3399.937 | 1.74 | 3324.829 | 3.91 | 3315.157 | 4.38 |
| 2012/11/1 | 3442.907 | 3389.608 | 1.55 | 3337.861 | 3.05 | 3349.827 | 2.78 |
| 2012/12/1 | 3460.208 | 3379.418 | 2.33 | 3336.503 | 3.58 | 3393.732 | 1.96 |
| 2013/1/1 | 3460.208 | 3406.014 | 1.57 | 3371.989 | 2.55 | 3427.727 | 0.95 |
| 2013/2/1 | 3477.509 | 3446.374 | 0.90 | 3410.133 | 1.94 | 3452.011 | 0.74 |
| 2013/3/1 | 3460.208 | 3462.169 | 0.06 | 3449.721 | 0.30 | 3482.668 | 0.64 |
| 2013/4/1 | 3494.81 | 3485.798 | 0.26 | 3502.356 | 0.22 | 3543.963 | 1.39 |
| 2013/5/1 | 3512.111 | 3524.423 | 0.35 | 3555.754 | 1.24 | 3625.738 | 3.13 |
| 2013/6/1 | 3512.111 | 3591.262 | 2.25 | 3684.274 | 4.90 | 3699.022 | 5.05 |
| 2013/7/1 | 3615.917 | 3695.444 | 2.20 | 3802.473 | 5.16 | 3738.225 | 3.27 |
| 2013/8/1 | 3719.723 | 3747.513 | 0.75 | 3830.496 | 2.98 | 3779.618 | 1.58 |
| 2013/9/1 | 3650.519 | 3740.002 | 2.45 | 3832.151 | 4.98 | 3825.664 | 4.58 |
| 2013/10/1 | 3685.121 | 3795.259 | 2.99 | 3877.587 | 5.22 | 3848.299 | 4.24 |

Table 4 Comparison between the predicted values of cumulative displacement and measured values at monitoring station ZG86

| Time | Measured value (mm) | GA-LSSVM | | GRNN | | BP | |
|---|---|---|---|---|---|---|---|
| | | Predicted value (mm) | Relative error (%) | Predicted value (mm) | Relative error (%) | Predicted value (mm) | Relative error (%) |
| 2012/10/1 | 4273.356 | 4183.984 | 2.09 | 4094.396 | 4.19 | 4096.849 | 4.13 |
| 2012/11/1 | 4290.657 | 4201.857 | 2.07 | 4104.93 | 4.33 | 4124.839 | 3.86 |
| 2012/12/1 | 4307.958 | 4149.796 | 3.67 | 4094.607 | 4.95 | 4091.602 | 5.02 |
| 2013/1/1 | 4307.958 | 4164.444 | 3.33 | 4130.77 | 4.11 | 4133.425 | 4.05 |
| 2013/2/1 | 4325.26 | 4192.775 | 3.06 | 4172.182 | 3.54 | 4186.816 | 3.20 |
| 2013/3/1 | 4342.561 | 4256.46 | 1.98 | 4212.082 | 3.00 | 4246.771 | 2.21 |
| 2013/4/1 | 4377.163 | 4317.892 | 1.35 | 4247.617 | 2.96 | 4312.109 | 1.49 |
| 2013/5/1 | 4394.464 | 4326.232 | 1.55 | 4297.626 | 2.20 | 4386.529 | 0.18 |
| 2013/6/1 | 4446.367 | 4388.693 | 1.30 | 4403.464 | 0.96 | 4523.094 | 1.73 |
| 2013/7/1 | 4532.872 | 4495.404 | 0.83 | 4535.948 | 0.07 | 4607.573 | 1.65 |
| 2013/8/1 | 4619.377 | 4609.902 | 0.21 | 4626.647 | 0.16 | 4656.543 | 0.80 |
| 2013/9/1 | 4602.076 | 4579.721 | 0.49 | 4628.676 | 0.58 | 4661.733 | 1.30 |
| 2013/10/1 | 4602.076 | 4592.204 | 0.21 | 4754.15 | 3.30 | 4713.128 | 2.41 |

Table 5 Comparison between the predicted values of cumulative displacement and measured values at monitoring station ZG87

| Time | Measured value (mm) | GA-LSSVM | | GRNN | | BP | |
|---|---|---|---|---|---|---|---|
| | | Predicted value (mm) | Relative error (%) | Predicted value (mm) | Relative error (%) | Predicted value (mm) | Relative error (%) |
| 2012/10/1 | 1505.19 | 1561.869 | 3.77 | 1578.221 | 4.85 | 1583.026 | 5.17 |
| 2012/11/1 | 1522.491 | 1580.602 | 3.82 | 1590.364 | 4.46 | 1597.352 | 4.92 |
| 2012/12/1 | 1522.491 | 1580.359 | 3.80 | 1586.605 | 4.21 | 1591.506 | 4.53 |
| 2013/1/1 | 1522.491 | 1581.923 | 3.90 | 1593.249 | 4.65 | 1600.855 | 5.15 |
| 2013/2/1 | 1539.792 | 1585.652 | 2.98 | 1599.822 | 3.90 | 1606.609 | 4.34 |
| 2013/3/1 | 1557.093 | 1600.959 | 2.82 | 1605.274 | 3.09 | 1617.769 | 3.90 |
| 2013/4/1 | 1557.093 | 1601.648 | 2.86 | 1606.713 | 3.19 | 1606.812 | 3.19 |
| 2013/5/1 | 1557.093 | 1608.744 | 3.32 | 1612.571 | 3.56 | 1615.702 | 3.76 |
| 2013/6/1 | 1574.394 | 1620.881 | 2.95 | 1622.897 | 3.08 | 1618.934 | 2.83 |
| 2013/7/1 | 1574.394 | 1620.703 | 2.94 | 1632.984 | 3.72 | 1623.904 | 3.14 |





| 2013/8/1 | 1591.696 | 1631.651 | 2.51 | 1643.08 | 3.23 | 1637.051 | 2.85 |
| 2013/9/1 | 1591.696 | 1630.511 | 2.44 | 1652.604 | 3.83 | 1647.566 | 3.51 |
| 2013/10/1 | 1591.696 | 1633.119 | 2.60 | 1653.808 | 3.90 | 1653.139 | 3.86 |

Fig. 13 Measured values versus predicted values of the cumulative displacement: (a) monitoring station ZG85, (b) monitoring
station ZG86, and (c) monitoring station ZG87
**5 Verification and error analyses**
Three loss functions are used to assess the prediction performance and accuracy of the proposed model: the root mean
square error (*RMSE*), mean absolute error (*MAE*), and mean absolute percentage error (*MAPE*). Then, the optimal parameters with
minimum error are used to train the LSSVM model. The *RMSE*, *MAE* and *MAPE* formulas are as follows:

$$RMSE = \sqrt{\frac{1}{n}\sum_{i=1}^{n}(s_i - s_i^*)^2} \qquad (17)$$

$$MAE = \frac{1}{n}\sum_{i=1}^{n}\left|s_i - s_i^*\right| \qquad (18)$$

$$MAPE = \frac{1}{n}\sum_{i=1}^{n}\left|\frac{s_i - s_i^*}{s_i}\right| \qquad (19)$$

where $s_i$ is the measured value, $s_i^*$ is the predicted value, and $n$ is the number of predicted values.





The performances of different models for landslide displacement prediction are assessed based on the *RMSE*, *MAE* and
*MAPE*, as presented in Table 6. The prediction precision of the GA-LSSVM model based on time series analysis is better than that
of the GRNN and the BP. Notably, the *RMSE*, *MAE* and *MAPE* values of the GA-LSSVM model were 63.4076, 56.6098 and
1.587% lower than those of the GRNN model, respectively, and 49.3696, 43.5537 and 1.225% lower than those of the BP model
for monitoring station ZG85. The predicted results for monitoring stations ZG86 and ZG87 exhibited similar trends. According to
the prediction results, the GA-LSSVM model has good deduction ability for landslide displacement prediction and can provide
assistance in early risk assessment and landslide forecasting.
Table 6 Comparison of the performance of cumulative displacement prediction for the three models

| Model | *RMSE* (mm) | | | *MAE* (mm) | | | *MAPE* (%) | | |
|---|---|---|---|---|---|---|---|---|---|
| | ZG85 | ZG86 | ZG87 | ZG85 | ZG86 | ZG87 | ZG85 | ZG86 | ZG87 |
| GA-LSSVM | 62.4146 | 87.7215 | 49.0485 | 53.0048 | 74.0601 | 48.5392 | 1.492 | 1.703 | 3.131 |
| GRNN | 125.8222 | 134.6764 | 59.8173 | 109.6146 | 115.1067 | 59.2756 | 3.079 | 2.643 | 3.821 |
| BP | 111.7842 | 123.1948 | 62.0223 | 96.5585 | 107.6724 | 60.9701 | 2.717 | 2.464 | 3.935 |

**6 Conclusion**
Landslide displacement prediction is one of the focuses of landslide research. In this paper, we use the deformation of a
step-like landslide (Shuping landslide) as an example. According to time series analysis, the cumulative displacement is
decomposed into a trend component displacement representing the trend of landslide deformation in the long term and a periodic
component displacement that represents short-term deformation fluctuations. The trend displacement and periodic displacement
are predicted using a polynomial function and the GA-LSSVM model, respectively. The LSSVM yields good fitting results in
predicting the periodic displacement with the GA, which is utilized to determine the optimal parameters of the LSSVM. Based on
our analysis of the deformation of Shuping landslide, the reservoir water level, rainfall and groundwater have major influences on
the cumulative displacement. Therefore, based on the relational grades, we select six influential factors as the input variables. The
predicted cumulative displacement is obtained from the sum of the predicted trend displacement and the predicted periodic
displacement.
The GA-LSSVM model displays the highest accuracy, the smallest *RMSE* of 62.4146 mm, the smallest *MAE* of 53.0048
mm, and the smallest *MAPE* of 1.492% at monitoring station ZG85, while these three values are 87.7215 mm, 74.0601 mm and
1.703% at monitoring station ZG86 and 49.0485 mm, 48.5392 mm and 3.131% at monitoring station ZG87. The study results
show that GA-LSSVM provides good performance for landslide displacement prediction, and the GA is appropriate for
determining the optimal parameters used in the LSSVM model. Thus, the GA-LSSVM model can be effectively used to predict
landslide displacement and reflect the corresponding relationships between the major influencing factors and the periodic
component displacement.
**Acknowledgments**
The authors would like to acknowledge gratefully the Editor and the anonymous reviewers for their constructive criticism
on the earlier version of this paper and offering valuable suggestions that contributed to its improvement. This study was
supported by National Natural Science Foundation of China, Key Project of National Science Foundation of China (No. 41230637,
No. 41502290), the Ministry of Science and Technology of the P. R. China, National Basic Research Program of China (973
Program) (2011CB710600).

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
