# Peer review of "Landslide displacement prediction using the GA-LSSVM model and time series analysis: a case study of Three Gorges Reservoir, China"

_Natural Hazards and Earth System Sciences, 2017_

## Short Comment (SC1) · 22 May 2017

This paper presents an interesting time series analysis, where the cumulative displacement is decomposed into trend component displacement representing the trend of the landslide deformation in the long-term and periodic component displacement exhibiting the short-term deformation fluctuation. However, this manuscript is suggested "accept after minor revisions," provided that the following considerations are addressed and the manuscript revised accordingly. 1.The reviewer suggests verification of or discussion on why "The trend displacement and the periodic displacement are predicted by polynomial function and the GA-LSSVM model, respectively." 2.The Section 3.2 (Monitoring data and deformation characteristics of the landslide) describes the monitoring

data used in this study. The reviewer suggests validate these sampling data for land-slide displacement prediction in the very beginning; i.e. are those continuous and mutually dependent landslide data applicable or feasible to the statistical method, such as the least squares support vector machine (LSSVM), basically dealing with indepen-dent sampling data. 3.In the text there are 8 monitoring stations with GPS, in figure 4 stations appear. The 8 stations only appear in figure 6, but in the work and in the other figures and in the tables only appear data of 3 GPS stations. 4.Abstract section should contain important numerical figures such as relative error or RMS error figure. 5.Numerical improvement in some circumstances may not be the core technology and sometimes it may be redundant, leading to the suggestion of redirection and emphasis on why we need such a model analysis in real applications perhaps included in the Introduction section.

---

## Referee Comment (RC1) · Anonymous Referee #1 · 1 Jun 2017

The authors develop a new model based approach to study the landslide displacement patterns of a slow-moving landslide at the Three Gorges Reservoir, China. The landslide movement rates are shown to influenced by a combination of both rainfall conditions and reservoir level fluctuations. The study uses monitoring records from the landslide to train and test the model before assessing its capability to predict future ground movements. The results illustrate that the model predictions and displacement behavior of the landslide are broadly consistent and therefore may provide a useful tool in the forecasting of future movements at the site. The study uses novel approaches and will be of interest to a broad readership of NHESS. It is recommended that the article be accepted for publication following revisions as suggested below. Further minor comments and suggestions are provided in the reviewed manuscript attached.

General: The landslide appears to show continuous ongoing slow-movement with periods of episodic accelerated ground creep associated with rainfall and reservoir level changes. The highest rates of movement observed in the landslide appear to occur following periods of heavy intense rainfall after the lake level has been reduced. This behavior is like to be best explained in terms of stress changes within the landslide in that the rainfall events cause increased pore water pressures in the landslide shear zone which reduces the effective stress and increases instability. Similarly, the lowering of the lake level reduces the confining stress whilst pore water pressures are still high which would promote accelerated movement. I would argue that this behavior is common in a number of slopes and therefore the explanation for the movement observed at this site should be considered in this context. Statements such as the degrading of the sliding mass by excess moisture and increases in sliding forces are therefore not likely to be the key driving mechanism in slope instability.

Introduction: Pg 1_line 28 - Geological conditions here are referred to as an external factor influencing landslides. Geoelogical conditions should be considered as an internal factor as is later suggested in the manuscript. Pg 1_line 31-35 - These final sentences of the first paragraph should really be the start of the introduction as this sets out the general motivation for the study before linking this to the site. Pg 2_lines 31-34 - I am not sure that this is needed I would suggest deleting this.

Methodology: Pg 2_ lines 38-39 – I'm not sure I fully understand this point. Landslide displacement is caused by both internal and external factors but why does the lithology, geological structure and topography cause result in monotonic displacement through time? Also groundwater (pore water pressures) should be considered here. Most likely the ground water table remains high enough for ongoing movement to continue. Pg 5_lines 3-16 – This section This section introduces the GA computational model but largely explains this through its previous biological applications. It would be much easier for the reader to explain how this has been adapted for landslide studies.

[Figure]

Case Study Pg 6_ line 29 – Why is landslide monitoring considered a qualitative approach to analyse landslide development. This is quantitative data. Pg 8_lines 17 – 18 – This should be the other way around- the landslide stability decreased and the deformation increased. Pg 9_line 24 - Statement 'materials in the sliding mass are degraded by excess moisture and additional hydrodynamic pressure' is not correct. The excess pore water pressure reduces the mean effective stress at the landslide shear surface making it more susceptible to movement. Pg 12_ lines9-10 – How has the sliding force increased? Is it not the case that the confining pressure reduces with the lowering of the lake but the pore water pressure remains high so this change in stress state makes the slope more unstable? Pg 12_line 12 - Is this an actual piezometer or standpipe installation or is this water observed within the inclinometer tube itself? If the latter is there any certainty as to where this has come from? If not an installed piezometer it could have come from the top cap of the installation and therefore may not be a reliable groundwater measurement.

Figures Fig 4. The key is not explained. A clear key showing instrument type and borehole locations is needed Fig 5. As with figure 4 the key is not clear. Also the borehole and inclinometers should be drawn on to show their depth. Fig 8. Diagrams are hard to read. It would be better to display these as conventional inclinometer plots with depth on the y axis and displacement on the x axis.

Please also note the supplement to this comment:
http://www.nat-hazards-earth-syst-sci-discuss.net/nhess-2017-87/nhess-2017-87-RC1-supplement.pdf
* * *
[Figure]

**Supplement:**

This is just a preview and not the published paper.

[revised manuscript text omitted]

---

## Author Comment (AC1) · 10 Jul 2017

Response to Reviewer's Comments The authors would like to thank the reviewer for the careful reading and constructive comments that have helped sharpen this manuscript. In this revision, all the comments of the reviewers have been carefully addressed. Specific responses to the review comments are listed below. The line numbers refer to those in the revised manuscript.

Anonymous Referee #1

1. Introduction: Pg 1_line 28 – Geological conditions here are referred to as an external

factor influencing landslides. Geological conditions should be considered as an internal factor as is later suggested in the manuscript.

Response: Thank you for the careful reading. Indeed, geological conditions are considered as an internal factor in the manuscript. We have revised the manuscript accordingly. Please see Introduction: Pg 1_lines 35-36 in this revision.

2. Introduction: Pg 1_line 31-35 – These final sentences of the first paragraph should really be the start of the introduction as this sets out the general motivation for the study before linking this to the site.

Response: Thank you for the careful reading and constructive comment. We agreed with the reviewer. We have revised the text accordingly. Please see Introduction: Pg 1_lines 28-33 in this revision.

3. Introduction: Pg 2_lines 31-34 – I am not sure that this is needed I would suggest deleting this.

Response: Thank you for the comment. In this submission, we have revised the text accordingly. Please see Introduction: Pg 2_lines 35-36 in this revision.

4. Methodology: Pg 2_ lines 38-39 – I'm not sure I fully understand this point. Landslide displacement is caused by both internal and external factors but why does the lithology, geological structure and topography cause result in monotonic displacement through time? Also groundwater (pore water pressures) should be considered here. Most likely the ground water table remains high enough for ongoing movement to continue.

Response: Thank you for the careful reading and kind comment. To avoid the potential confusion, we have revised the text accordingly, with slight modification. Please see 2.1 Time series analysis of displacement: Pg 2_lines 41-46 in this revision.

The nonlinear evolution process of the cumulative displacement of landslides is controlled by primary factors such as geological conditions, and trigger factors such as rainfall and reservoir water level changes. The displacement of landslide sequence is

an instability time series. Based on the time series analysis, total displacement of land-slide can be broken down into different corresponding components according to the different influential factors. Total displacement of landslide can be divided into trend component displacement, which is affected by the periodic dynamic functioning of inducing factors such as rainfall, reservoir water level, groundwater. Trend component displacement nearly increases under large time scales, and periodic component displacement fluctuated increases under small time scales. The trend component revealed the long-term trend of the sequence, which is determined by the potential energy and constraint condition of the slope. Many landslides exhibit long-lasting, continuous movements under gravity loads that are affected by the creep property of slope materials (Desai et al. 1995). One of the important factors that influence the behavior of creeping slopes is appropriate characterization of the response of geologic materials and interfaces; in the case of creeping slopes, the latter can occur at the junction of the creeping mass and the essentially stationary (rock) mass below it. Landslide deformation is often characterized by creep, which generally need to undergo three stages, initial deformation, stable deformation and accelerated deformation stage. In the evolution scheme of three deformation phases of landslide, the landslide displacement generally increases monotonically with time.

Furthermore, we agreed with the reviewer that groundwater (pore water pressures) should be considered here. Groundwater, which is regarded as an active geologic agent, is one of the main factors that induces landslide instability. In the rising phase of reservoir water level, the groundwater level gradually increases, with a slight lag behind the increase in the reservoir water level. The groundwater remains high enough for ongoing movement to continue.

5. Methodology: Pg 5_lines 3-16 – This section This section introduces the GA computational model but largely explains this through its previous biological applications. It would be much easier for the reader to explain how this has been adapted for landslide studies.

Response: Thank you for the kind comment. We have revised the text accordingly.

We agreed with the reviewer that this section is not relevant to landslide, just introduces the GA model from the biological point of view. Thus, we deleted this section in the revision. About how this has been adapted for landslide studies, we had explained it in the introduction: Pg 2_lines 25-27.

6. Case Study: Pg 6_ line 29 – Why is landslide monitoring considered a qualitative approach to analyse landslide development. This is quantitative data.

Response: Thank you for the careful reading and kind comment. We have revised the text accordingly. Indeed, landslide monitoring is considered a qualitative approach to analyse landslide development. We are very sorry for that our mistake in spelling words. Please see Case Study: Pg 6_line 28 in this revision.

7. Case Study: Pg 8_lines 17-18 – This should be the other way around- the landslide stability decreased and the deformation increased.

Response: Thank you for the careful reading and kind comment. We have revised the text accordingly. Please see Case Study: Pg 8_line 20-21 in this revision.

8. Case Study: Pg 9_line 24 – Statement 'materials in the sliding mass are degraded by excess moisture and additional hydrodynamic pressure' is not correct. The excess pore water pressure reduces the mean effective stress at the landslide shear surface making it more susceptible to movement.

Response: Thank you for the careful reading and constructive comments. We agreed with the reviewer about the explanation. We have revised the text accordingly. Please see Case Study: Pg 12_line 23-24 in this revision.

9. Case Study: Pg 12_lines9-10 – How has the sliding force increased? Is it not the case that the confining pressure reduces with the lowering of the lake but the pore water pressure remains high so this change in stress state makes the slope more unstable?

Response: Thank you for the constructive comments. We have revised the text accordingly. Please see Case Study: Pg 12_line 5-13 in this revision for details.

Although the variation in reservoir water level was small before April 2007, the periodic displacement still exhibited small fluctuations due to the effects of rainfall and groundwater. This behavior could be explained in terms of stress changes within the landslide in that the rainfall events cause increased pore water pressures in the landslide shear zone which reduced the effective stress and increased instability. After April 2007, several distinct peaks can be observed in the periodic displacement-time curves during periods of decreasing reservoir water level. For example, the periodic displacement increased from May to July 2009 and from May to September 2012. However, when the reservoir water level increased from 145 m to 175 m, the periodic displacement gradually decreased. The main reason for the above conditions was that the rise of the reservoir water level increased the confining stress on the surface of the landslide and the hydrodynamic pressure, the direction of which was toward the interior of sliding body. Similarly, the lowering of the reservoir water level reduced the confining stress whilst pore water pressures were still high which would promote accelerated movement.

10. Case Study: Pg 12_line 12 – Is this an actual piezometer or standpipe installation or is this water observed within the inclinometer tube itself? If the latter is there any certainty as to where this has come from? If not an installed piezometer it could have come from the top cap of the installation and therefore may not be a reliable groundwater measurement.

Response: Thank you for the careful reading and constructive comments. We have revised the text accordingly. Please see Case Study: Pg 12_line 15-16 in this revision for details.

The water gauge used in this landslide was 730 type water level sensor, with the characteristics of measuring range as deep as 210 meters, high measuring accurancy and

stable performance. The data acquisition and memory used NetL G-301 data storage device. At the head scarp of the landslide at an elevation of 181m, groundwater depth was measured by water gauge within inclinometer monitoring hole QZK3.

11. Figures Fig 4. The key is not explained. A clear key showing instrument type and borehole locations is needed Fig 5. As with figure 4 the key is not clear. Also the borehole and inclinometers should be drawn on to show their depth. Fig 8. Diagrams are hard to read. It would be better to display these as conventional inclinometer plots with depth on the y axis and displacement on the x axis.

Response: Thank you for the careful reading and constructive comments. We have revised the text accordingly.

About Fig. 4 and Fig.5, we have added the key for legend information accordingly. Please see Fig.4 and Fig.5 in this revision for details.

Furthermore, Fig 8. Diagrams is displayed as conventional inclinometer plots with depth on the y axis and displacement on the x axis. Please see Fig 8 in this revision.

References Furuya, G., Sassa, K., Hiura, H., Fukuoka, H.: Mechanism of creep movement caused by landslide activity and underground erosion in crystalline schist, Shikoku Island, southwestern Japan, Eng Geol, 53, 311-325, 10.1016/S0013-7952(98)00084-2, 1999. Desai, C. S., Samtani, N. C., Vulliet, L.: Constitutive modeling and analysis of creeping slopes, J Geotech Eng Trans ASCE, 121,43-56, 10.1061/(ASCE)0733-9410(1995)121:1(43), 1995. Sun, M., Tang, H., Wang, M., Shan, Z., Hu X.: Creep behavior of slip zone soil of the Majiagou landslide in the Three Gorges area, Environ Earth Sci, 16, 1-12, 10.1007/s12665-016-6002-x, 2016. Haq, A. N., Marimuthu, P., Jeyapaul, R.: Multi response optimization of machining parameters of drilling Al/SiC metal matrix composite using grey relational analysis in the Taguchi method, The Int J Adv Manuf Technol, 37, 250-255, 10.1007/s00170-007-0981-4, 2008.

Please also note the supplement to this comment:
https://www.nat-hazards-earth-syst-sci-discuss.net/nhess-2017-87/nhess-2017-87-AC1-supplement.pdf

**Supplement:**

**Response to Reviewer's Comments**

The authors would like to thank the reviewer for the careful reading and constructive comments that have helped sharpen this manuscript. In this revision, all the comments of the reviewers have been carefully addressed. Specific responses to the review comments are listed below. The line numbers refer to those in the revised manuscript.

**Anonymous Referee #1**

*1. Introduction: Pg 1_line 28 – Geological conditions here are referred to as an external factor influencing landslides. Geological conditions should be considered as an internal factor as is later suggested in the manuscript.*

Response: Thank you for the careful reading. Indeed, geological conditions are considered as an internal factor in the manuscript. We have revised the manuscript accordingly. Please see **Introduction: Pg 1_lines 35-36** in this revision.

*2. Introduction: Pg 1_line 31-35 – These final sentences of the first paragraph should really be the start of the introduction as this sets out the general motivation for the study before linking this to the site.*

Response: Thank you for the careful reading and constructive comment. We agreed with the reviewer. We have revised the text accordingly. Please see **Introduction: Pg 1_lines 28-33** in this revision.

*3. Introduction: Pg 2_lines 31-34 – I am not sure that this is needed I would suggest deleting this.*

Response: Thank you for the comment. In this submission, we have revised the text accordingly. Please see **Introduction: Pg 2_lines 35-36** in this revision.

*4. Methodology: Pg 2_ lines 38-39 – I'm not sure I fully understand this point. Landslide displacement is caused by both internal and external factors but why does the lithology, geological structure and topography cause result in monotonic displacement through time? Also groundwater (pore water pressures) should be considered here. Most likely the ground water table remains high enough for ongoing movement to continue.*

Response: Thank you for the careful reading and kind comment. To avoid the potential confusion, we have revised the text accordingly, with slight modification. Please see **2.1 Time series analysis of displacement: Pg 2_lines 41-46** in this revision.

The nonlinear evolution process of the cumulative displacement of landslides is controlled by primary factors such as geological conditions, and trigger factors such as rainfall and reservoir water level changes. The displacement of landslide sequence is an instability time series. Based on the time series analysis, total displacement of landslide can be broken down into different corresponding components according to the different influential factors. Total displacement of landslide can be divided into trend

component displacement, which is affected by the periodic dynamic functioning of inducing factors such as rainfall, reservoir water level, groundwater. Trend component displacement nearly increases under large time scales, and periodic component displacement fluctuated increases under small time scales. The trend component revealed the long-term trend of the sequence, which is determined by the potential energy and constraint condition of the slope.

Many landslides exhibit long-lasting, continuous movements under gravity loads that are affected by the creep property of slope materials (Desai et al. 1995). One of the important factors that influence the behavior of creeping slopes is appropriate characterization of the response of geologic materials and interfaces; in the case of creeping slopes, the latter can occur at the junction of the creeping mass and the essentially stationary (rock) mass below it. Landslide deformation is often characterized by creep, which generally need to undergo three stages, initial deformation, stable deformation and accelerated deformation stage. In the evolution scheme of three deformation phases of landslide, the landslide displacement generally increases monotonically with time.

Furthermore, we agreed with the reviewer that groundwater (pore water pressures) should be considered here. Groundwater, which is regarded as an active geologic agent, is one of the main factors that induces landslide instability. In the rising phase of reservoir water level, the groundwater level gradually increases, with a slight lag behind the increase in the reservoir water level. The groundwater remains high enough for ongoing movement to continue.

*5. Methodology: Pg 5_lines 3-16 – This section This section introduces the GA computational model but largely explains this through its previous biological applications. It would be much easier for the reader to explain how this has been adapted for landslide studies.*

Response: Thank you for the kind comment. We have revised the text accordingly.

We agreed with the reviewer that this section is not relevant to landslide, just introduces the GA model from the biological point of view. Thus, we deleted this section in the revision. About how this has been adapted for landslide studies, we had explained it in the introduction: Pg 2_lines 25-27.

*6. Case Study: Pg 6_ line 29 – Why is landslide monitoring considered a qualitative approach to analyse landslide development. This is quantitative data.*

Response: Thank you for the careful reading and kind comment. We have revised the text accordingly. Indeed, landslide monitoring is considered a qualitative approach to analyse landslide development. We are very sorry for that our mistake in spelling words. Please see **Case Study: Pg 6_line 28** in this revision.

*7. Case Study: Pg 8_lines 17-18 – This should be the other way around- the landslide stability decreased and the deformation increased.*

Response: Thank you for the careful reading and kind comment. We have revised the text accordingly. Please see **Case Study: Pg 8_line 20-21** in this revision.

*8. Case Study: Pg 9_line 24 – Statement 'materials in the sliding mass are degraded by excess moisture and additional hydrodynamic pressure' is not correct. The excess pore water pressure reduces the mean effective stress at the landslide shear surface making it more susceptible to movement.*

Response: Thank you for the careful reading and constructive comments. We agreed with the reviewer about the explanation. We have revised the text accordingly. Please see **Case Study: Pg 12_line 23-24** in this revision.

*9. Case Study: Pg 12_lines9-10 – How has the sliding force increased? Is it not the case that the confining pressure reduces with the lowering of the lake but the pore water pressure remains high so this change in stress state makes the slope more unstable?*

Response: Thank you for the constructive comments. We have revised the text accordingly. Please see **Case Study: Pg 12_line 5-13** in this revision for details.

Although the variation in reservoir water level was small before April 2007, the periodic displacement still exhibited small fluctuations due to the effects of rainfall and groundwater. This behavior could be explained in terms of stress changes within the landslide in that the rainfall events cause increased pore water pressures in the landslide shear zone which reduced the effective stress and increased instability. After April 2007, several distinct peaks can be observed in the periodic displacement-time curves during periods of decreasing reservoir water level. For example, the periodic displacement increased from May to July 2009 and from May to September 2012. However, when the reservoir water level increased from 145 m to 175 m, the periodic displacement gradually decreased. The main reason for the above conditions was that the rise of the reservoir water level increased the confining stress on the surface of the landslide and the hydrodynamic pressure, the direction of which was toward the interior of sliding body. Similarly, the lowering of the reservoir water level reduced the confining stress whilst pore water pressures were still high which would promote accelerated movement.

*10. Case Study: Pg 12_line 12 – Is this an actual piezometer or standpipe installation or is this water observed within the inclinometer tube itself? If the latter is there any certainty as to where this has come from? If not an installed piezometer it could have come from the top cap of the installation and therefore may not be a reliable groundwater measurement.*

Response: Thank you for the careful reading and constructive comments. We have revised the text accordingly. Please see **Case Study: Pg 12_line 15-16** in this revision for details.

The water gauge used in this landslide was 730 type water level sensor, with the characteristics of measuring range as deep as 210 meters, high measuring accurancy and stable performance. The data acquisition and memory used NetL G-301 data storage device. At the head scarp of the landslide at an elevation of 181m, groundwater depth was measured by water gauge within inclinometer monitoring hole QZK3.

*11. Figures Fig 4. The key is not explained. A clear key showing instrument type and borehole*

*locations is needed Fig 5. As with figure 4 the key is not clear. Also the borehole and inclinometers should be drawn on to show their depth. Fig 8. Diagrams are hard to read. It would be better to display these as conventional inclinometer plots with depth on the y axis and displacement on the x axis.*

Response: Thank you for the careful reading and constructive comments. We have revised the text accordingly.

About Fig. 4 and Fig.5, we have added the key for legend information accordingly. Please see **Fig.4 and Fig.5** in this revision for details.

Furthermore, Fig 8. Diagrams is displayed as conventional inclinometer plots with depth on the *y* axis and displacement on the *x* axis. Please see **Fig 8** in this revision.

[revised manuscript text omitted]

---

## Author Comment (AC2) · 10 Jul 2017

Response to Interactive Comments The authors thank the reviewer and the Editor for their comments. We have addressed all the comments in the revised manuscript. The following includes our point-by-point responses to the comments and the locations in the manuscript where the corresponding revisions appear.

Interactive Comments

1.The reviewer suggests verification of or discussion on why "The trend displacement and the periodic displacement are predicted by polynomial function and the GA-LSSVM

model, respectively."

Response: Thank you for the careful reading and constructive comment. We have revised the text accordingly. Please see Methodology: Pg 2_line 37-46 in this revision.

Cumulative displacement of landslides is caused by the combined effects of internal geological conditions (lithology, geological structure, topography, etc.) and external environmental factors (rainfall, reservoir water level, groundwater, etc.). The landslide displacement caused by internal geological conditions increases generally with time, which reflects the trend in cumulative displacement. Landslide deformation exhibit long-lasting and continuous movements under gravity loads that is affected by the creep characteristic. Geological structure and topography cause result in monotonic displacement through time. Because the curves of the trend component displacement versus time have quasi-linear and incremental characteristics, we use polynomial functions to fit these curves and provide the best-fitted results. However, the landslide displacement induced by external environmental factors is approximately periodic. The LSSVM model yields good performance in pattern recognition and nonlinear function fitting. The genetic algorithm (GA) is a global optimization algorithm that uses highly parallel, random and adaptive searching based on biological natural selection and optimization. In this paper, the GA is selected as the method of parameter optimization in the LSSVM due to its advantages in determining the unknown parameters that are consistent between the predicted data and the measured data. By introducing the GA, some key parameters of the LSSVM model can be derived automatically. Therefore, considering the characteristics of the periodic displacement and the relational grades between variables, we select the combination of the LSSVM model and the GA to predict landslide periodic displacement.

2.The Section 3.2 (Monitoring data and deformation characteristics of the landslide) describes the monitoring data used in this study. The reviewer suggests validate these sampling data for landslide displacement prediction in the very beginning; i.e. are those continuous and mutually dependent landslide data applicable or feasible to the statistical method, such as the least squares support vector machine (LSSVM), basically dealing with independent sampling data.

Response: Thank you for the careful reading and constructive comment. We have revised the text accordingly. Please see Introduction: Pg 1_line 31-32 and line 38-39 in this revision. It is well known that the evolution process of landslide is a complex non-linear process that is caused by the complex interaction of different factors, e.g. the complicated geological settings, varying hydrological conditions. Displacement time series are generally appreciated as the direct representation of the complex and non-linear dynamical behaviour of landslide. However, the landslide displacement induced by the exteral factors is approximately periodic. Therefore, a landslide displacement sequence is an instability time series with a periodic episodic movement characteristic. Because the integrity of the data collected at monitoring points has an effect on the displacement prediction, the monitoring data from July 2003 to October 2013 are selected to explore landslide deformation. The SVM is a machine learning model based on the knowledge of statistical learning for small samples and structural risk minimization. With the rapid development of theory and technique, Least Squares Support Vector Machines (LSSVM) have been proposed for overcoming the defects of the SVM with high computational complexity due to quadratic programming. Compared with SVM, LSSVM runs faster and exhibits more adaptability because the quadratic optimization problem of SVM is transformed into a linear system of equations, and it has been widely used in the automatic and efficient monitoring of landslide safety.

3.In the text there are 8 monitoring stations with GPS, in figure 4 stations appear. The 8 stations only appear in figure 6, but in the work and in the other figures and in the tables only appear data of 3 GPS stations.

Response: Thank you for the comment. In this submission, about the reason for exhibiting 3 GPS stations in the work and in the other figures and in the tables were provided. Please see Landslide displacement prediction: Pg 11_Line 3-5.

In the text there are 8 monitoring stations with GPS, in figure 4 and figure 6 stations appear. 8 monitoring stations were used to analysed deformation characteristics of the landslide. But in the work and in the other figures and in the tables only appear data of 3 GPS stations, GP85, GP86, GP87. Because ZG85 in the head scarp areas, ZG86 in the middle areas and ZG87 in the back scarp areas were all from longitudinal section A-B and they represented different regions. Thus, due to the limited space, we just selected 3 GPS stations to validate the predicted performance of the proposed model. Based on the analysis of the deformation characteristics of Shuping landslide and the GA-LSSVM model and due to the obvious nonlinear and episodic movement deformation characteristics of monitoring stations ZG85, ZG86 and ZG87, we select only these stations along longitudinal section A-B to verify and establish the prediction model.

4.Abstract section should contain important numerical figures such as relative error or RMS error figure.

Response: Thank you for the comment. In this submission, some important numerical figures were provided. Please see Abstract section.

5.Numerical improvement in some circumstances may not be the core technology and sometimes it may be redundant, leading to the suggestion of redirection and emphasis on why we need such a model analysis in real applications perhaps included in the Introduction section.

Response: Thank you for the careful reading and constructive comment. We have revised the text accordingly. Please see Introduction.

As is well known, it is difficult to predict the displacement of a landslide accurately using a mathematical model. This is mainly because the landslide is characterized by complex nonlinear-dynamic behavior involving many uncertain geological and engineering factors. Recently, numerous models have been proposed and widely used for landslide displacement, such as functional regression, Artificial Neural Network

(ANN), and Support Vector Machines (SVMs). All those models tried to find the complex non-linear relationship between a training set of input vectors and corresponding output. However, ANN has its own drawbacks such as arriving at the local minimum, over fitting, slow convergence speed that limit its predictive performance. The SVM is a machine learning model based on the knowledge of statistical learning for small samples and structural risk minimization. Therefore, SVM becomes a more advanced method for dealing with the nonlinear problems in predicting landslide displacement. With the rapid development of theory and technique, Least Squares Support Vector machines (LSSVM) have been proposed for overcoming the defects of the SVM with high computational complexity due to quadratic programming. To improve the predictive performance, Genetic Algorithm (GA) was introduced to optimize the parameters of model for obtaining better predictive performance in recent achievements. Therefore, the present paper proposes a landslide displacement prediction model based on the GA-LSSVM with time series analysis.

Please also note the supplement to this comment:
https://www.nat-hazards-earth-syst-sci-discuss.net/nhess-2017-87/nhess-2017-87-AC2-supplement.pdf

**Supplement:**

**Response to Interactive Comments**

The authors thank the reviewer and the Editor for their comments. We have addressed all the comments in the revised manuscript. The following includes our point-by-point responses to the comments and the locations in the manuscript where the corresponding revisions appear.

**Interactive Comments**

*1.The reviewer suggests verification of or discussion on why "The trend displacement and the periodic displacement are predicted by polynomial function and the GA-LSSVM model, respectively."*

Response: Thank you for the careful reading and constructive comment. We have revised the text accordingly. Please see **Methodology: Pg 2_line 37-46** in this revision.

Cumulative displacement of landslides is caused by the combined effects of internal geological conditions (lithology, geological structure, topography, etc.) and external environmental factors (rainfall, reservoir water level, groundwater, etc.). The landslide displacement caused by internal geological conditions increases generally with time, which reflects the trend in cumulative displacement. Landslide deformation exhibit long-lasting and continuous movements under gravity loads that is affected by the creep characteristic. Geological structure and topography cause result in monotonic displacement through time. Because the curves of the trend component displacement versus time have quasi-linear and incremental characteristics, we use polynomial functions to fit these curves and provide the best-fitted results.

However, the landslide displacement induced by external environmental factors is approximately periodic. The LSSVM model yields good performance in pattern recognition and nonlinear function fitting. The genetic algorithm (GA) is a global optimization algorithm that uses highly parallel, random and adaptive searching based on biological natural selection and optimization. In this paper, the GA is selected as the method of parameter optimization in the LSSVM due to its advantages in determining the unknown parameters that are consistent between the predicted data and the measured data. By introducing the GA, some key parameters of the LSSVM model can be derived automatically. Therefore, considering the characteristics of the periodic displacement and the relational grades between variables, we select the combination of the LSSVM model and the GA to predict landslide periodic displacement.

*2.The Section 3.2 (Monitoring data and deformation characteristics of the landslide) describes the monitoring data used in this study. The reviewer suggests validate these sampling data for landslide displacement prediction in the very beginning; i.e. are those continuous and mutually dependent landslide data applicable or feasible to the statistical method, such as the least squares support vector machine (LSSVM), basically dealing with independent sampling data.*

Response: Thank you for the careful reading and constructive comment. We have revised the text accordingly. Please see **Introduction: Pg 1_line 31-32 and line 38-39** in this revision.

It is well known that the evolution process of landslide is a complex non-linear process that is caused by the complex interaction of different factors, e.g. the complicated geological settings, varying hydrological conditions. Displacement time series are generally appreciated as the direct representation of the complex and non-linear dynamical behaviour of landslide. However, the landslide displacement induced by the exteral factors is approximately periodic. Therefore, a landslide displacement sequence is an instability time series with a periodic episodic movement characteristic. Because the integrity of the data collected at monitoring points has an effect on the displacement prediction, the monitoring data from July 2003 to October 2013 are selected to explore landslide deformation.

The SVM is a machine learning model based on the knowledge of statistical learning for small samples and structural risk minimization. With the rapid development of theory and technique, Least Squares Support Vector Machines (LSSVM) have been proposed for overcoming the defects of the SVM with high computational complexity due to quadratic programming. Compared with SVM, LSSVM runs faster and exhibits more adaptability because the quadratic optimization problem of SVM is transformed into a linear system of equations, and it has been widely used in the automatic and efficient monitoring of landslide safety.

*3.In the text there are 8 monitoring stations with GPS, in figure 4 stations appear. The 8 stations only appear in figure 6, but in the work and in the other figures and in the tables only appear data of 3 GPS stations.*

Response: Thank you for the comment. In this submission, about the reason for exhibiting 3 GPS stations in the work and in the other figures and in the tables were provided. Please see **Landslide displacement prediction: Pg 11_Line 3-5**.

In the text there are 8 monitoring stations with GPS, in figure 4 and figure 6 stations appear. 8 monitoring stations were used to analysed deformation characteristics of the landslide. But in the work and in the other figures and in the tables only appear data of 3 GPS stations, GP85, GP86, GP87. Because ZG85 in the head scarp areas, ZG86 in the middle areas and ZG87 in the back scarp areas were all from longitudinal section A-B and they represented different regions. Thus, due to the limited space, we just selected 3 GPS stations to validate the predicted performance of the proposed model.

Based on the analysis of the deformation characteristics of Shuping landslide and the GA-LSSVM model and due to the obvious nonlinear and episodic movement deformation characteristics of monitoring stations ZG85, ZG86 and ZG87, we select only these stations along longitudinal section A-B to verify and establish the prediction model.

*4.Abstract section should contain important numerical figures such as relative error or RMS error figure.*

Response: Thank you for the comment. In this submission, some important numerical figures were provided. Please see **Abstract** section.

*5.Numerical improvement in some circumstances may not be the core technology and sometimes it may be redundant, leading to the suggestion of redirection and emphasis on why we need such a model analysis in real applications perhaps included in the Introduction section.*

Response: Thank you for the careful reading and constructive comment. We have revised the text accordingly. Please see **Introduction**.

As is well known, it is difficult to predict the displacement of a landslide accurately using a mathematical model. This is mainly because the landslide is characterized by complex nonlinear-dynamic behavior involving many uncertain geological and engineering factors.

Recently, numerous models have been proposed and widely used for landslide displacement, such as functional regression, Artificial Neural Network (ANN), and Support Vector Machines (SVMs). All those models tried to find the complex non-linear relationship between a training set of input vectors and corresponding output. However, ANN has its own drawbacks such as arriving at the local minimum, over fitting, slow convergence speed that limit its predictive performance. The SVM is a machine learning model based on the knowledge of statistical learning for small samples and structural risk minimization. Therefore, SVM becomes a more advanced method for dealing with the nonlinear problems in predicting landslide displacement. With the rapid development of theory and technique, Least Squares Support Vector machines (LSSVM) have been proposed for overcoming the defects of the SVM with high computational complexity due to quadratic programming. To improve the predictive performance, Genetic Algorithm (GA) was introduced to optimize the parameters of model for obtaining better predictive performance in recent achievements. Therefore, the present paper proposes a landslide displacement prediction model based on the GA-LSSVM with time series analysis.

[revised manuscript text omitted]

---

## Short Comment (SC2) · 19 Jul 2017

I think that the author has revised the manuscript carefully. In view of this, this article can be published. This paper illustrates that the prediction results and deformation activities of the landslide are basicly consistent. Thus, the model may provide a useful tool in the landslide displacement prediction at the site.

---

## Referee Comment (RC2) · Anonymous Referee #1 · 21 Jul 2017

The authors have attended to the comments provided. IT is my view that the paper is now suitable for publication.

---

## Referee Comment (RC3) · Anonymous Referee #2 · 28 Sep 2017

GA-LSSVM model and time series analysis were adopted in this paper to produce landslide displacement prediction. The results illustrate GA-LSSVM model can be effectively used to predict landslide displacement and have better predictive ability than GRNN model and BP model. Therefore, this study proved the reliability of this prediction method and spread it in an efficient way. This would be potentially interesting for the journal NHESS. However, there are still some spelling and grammatical mistakes need to be revise, some confusing expressions need to be improve and some questions need to be answer and explain. Thus, it is recommended that the article can be accepted for publication if major revisions can be made as follow suggestions. 1. Introduction P1_29,". . .external factors, such as geological conditions. . ."ïïjŇ geological

conditions should be internal factors. P1_36,"in recently years" should be " in recent years" P2_23-26, Here, the studies, which also used GA-LSSVM model to predict landslide displacement, are suggested to be mentioned. For example, Cai Z, Xu W, Meng Y, et al. Prediction of landslide displacement based on GA-LSSVM with multiple factors. Bulletin of Engineering Geology & the Environment, 2016, 75(2):637-646.

2. Methodology P3_24, "By searching or a function..."ïijŇhere "or" I guess is a spelling mistake. P4_17-18, I suggest the authors to supplement an equation contains both C and $\sigma$ to express the model. P4_22-P5_2, These could be mentioned in introduction or put forward in a discussion section. P5_23-24, "The sampling...sampling data." This sentence is confusing. Why the data is independent. P5_26, It is not strict to conclude GA-LSSVM model has higher accuracy than other models due to the consideration of the trigger factors. Some other models also consider the trigger factors. P6, Fig 2, The technical route of left part is not clear. The methodology section is too long, authors are suggested to focus the introduction on what is new and what is developed by the authors to use the methodology to predict landslide displacement.

3.Case study P6_7,P6_17,P6_20-21,P6_25,P6_27-28, language should be improved. P7, Fig.4,& P8, Fig.5,the numbers in the legend needs to be explained. P8_9-10, "in frontal area were relatively low" and "in the middle-rear areas were very high" are not consist with the monitoring data. P9_5, The location of the local road in fig.7 is suggested to be marked on the map P9_17-21, There is no groundwater monitoring method or data mention Here.

4.Landslide displacement prediction P11_5-6, "The model... regarding...", language should be improved. P11_17-19, R2 are calculated according to the total data or to the predictive part of the data? Fig.9 is suggested to mark the R2 ,calculated according to the predictive part of the data, on the curves. P12_11-16, " slight lag" is not described clearly. P12 Fig.10, P13 Fig.11, why the authors choose the current month and past two month as two time periods for the indexes of variation of reservoir water level and rainfall? Is this choice reasonable? Because the influence period
should be determined by detailed analyzing on the respond relationship between landslide displacement and influence factors. P13_16-18," the cumulative rainfall in the current month, the cumulative rainfall in the past two months, the reservoir water level, the variation in the reservoir water level in the current month, the variation in the reservoir water level in the past two months, and groundwater depth are selected as input variables", these variables have strong correlation, for instance, the reservoir water level and groundwater depth. Will this kind of dependent relationship between the variables influence the accuracy of prediction? How the authors think about it? P14_9-10,"Notably,…water level." However, Fig.12 (b) did not match well. P14-16 Table 3, Table 4, Table 5, the measured cumulative displacement data are not from small to large in time. For example, ZG85ïïjŇ2012/11/1, 3442.907mm is smaller than 2012/10/1, 3460.208mm. Please explain why the cumulative displacement decreased?

Please also note the supplement to this comment:
https://www.nat-hazards-earth-syst-sci-discuss.net/nhess-2017-87/nhess-2017-87-RC3-supplement.pdf

**Supplement:**

[revised manuscript text omitted]

---

## Author Comment (AC3) · 13 Oct 2017

The authors would like to thank the reviewer for the careful reading and constructive comments that have helped sharpen this manuscript. In this revision, all the comments of the reviewers have been carefully addressed. Specific responses to the review comments are listed below. The line numbers refer to those in the revised manuscript.

Anonymous Referee #2

1. Introduction P1_29, "...external factors, such as geological conditions..."ïijN geological C1 conditions should be internal factors.

[Figure]

Response: Thank you for the careful reading. Indeed, geological conditions are considered as an internal factor in the manuscript. We have revised the manuscript accordingly. Please see Introduction: Pg 1_lines 35-36 in this revision.

2. Introduction P1_36,"in recently years" should be " in recent years".

Response: Thank you for the careful reading. We have revised the text accordingly. Please see Introduction: Pg 1_lines 40 in this revision.

3. Introduction P2_23-26, Here, the studies, which also used GA-LSSVM model to predict landslide displacement, are suggested to be mentioned. For example, Cai Z, Xu W, Meng Y, et al. Prediction of landslide displacement based on GA-LSSVM with multiple factors. Bulletin of Engineering Geology & the Environment, 2016, 75(2):637-646.

Response: Thank you for the constructive comments. We have revised the text accordingly. Please see Introduction: Pg 2_line 34 in this revision.

4. Methodology P3_24, "By searching or a function. . ."ïijŃhere "or" I guess is a spelling mistake.

Response: Thank you for the careful reading. That is really a spelling mistake. We are very sorry for that our mistake in spelling words. We have revised the text accordingly. Please see Methodology: Pg 3_line 34 in this revision.

5. Methodology P4_17-18, I suggest the authors to supplement an equation contains both C and $\sigma$ to express the model.

Response: Thank you for the careful reading and constructive comments. We have revised the text accordingly. Please see Methodology: Pg 5_line 1-2 in this revision.

It can be seen from this paper that C is a penalty factor representing the penalty degree of the training samples, and is a parameter of the kernel function. The parameter of the model C and the parameter of the kernel function significantly influence the prediction performance. The parameter C represents the error tolerance. The more accurate the parameter is, the higher the prediction performance is, but this can lead to overtraining. The parameter implicitly determines the spatial distribution of data mapping in the new feature space. In this paper, the radial kernel function is selected as the kernel function in the LSSVM model, so the determination of the parameters C and $\sigma$ is very significant for the great prediction performance. However, the equation between C and $\sigma$ expressed jointly the Eq. (5) $\sim$ (10) is extremely complicated, which is inconvenient to be expressed by a certain formula. In the machine learning of LSSVM, the parameters C and $\sigma$ are hyper parameters that set the values before the beginning of the learning process, rather than the parameter data obtained by training. In general, it is necessary to optimize the hyper parameters and select a set of optimal hyper parameters for the machine learning of LSSVM to improve the performance and effective of learning. In this paper, the GA is selected as the method of parameter optimization in the LSSVM due to its advantages in determining the unknown parameters that are consistent between the predicted data and the measured data. By introducing the GA, the parameters C and $\sigma$ can be derived automatically. In the process of calculation, the best parameter C is first obtained by searching the optimum. Then based on the best parameter C, the best parameter $\sigma$ is obtained by training.

6. Methodology P4_22-P5_2, These could be mentioned in introduction or put forward in a discussion section.

Response: Thank you for the careful reading and kind comment. We have revised the text accordingly. Please see Introduction: Pg 2_line 24-31 in this revision.

7. Methodology P5_23-24, "The sampling...sampling data." This sentence is confusing. Why the data is independent.

Response: Thank you for the careful reading and constructive comment. We have revised the text accordingly. Indeed, the sampling data used for landslide displacement prediction are continuous and mutually dependent landslide data which are applicable or feasible to the specific method. We are very sorry for that our mistake in this sentence. Please see Introduction: Pg 3_line 1-6 and Methodology: Pg 5_line 25 in this revision.

It is well known that the evolution process of landslide is a complex non-linear process that is caused by the complex interaction of different factors, e.g. the complicated geological settings, varying hydrological conditions. Displacement time series are generally appreciated as the direct representation of the complex and non-linear dynamical behaviour of landslide. However, the landslide displacement induced by the exteral factors is approximately periodic. Therefore, a landslide displacement sequence is an instability time series with a periodic episodic movement characteristic. Because the integrity of the data collected at monitoring points has an effect on the displacement prediction, the monitoring data from July 2003 to October 2013 are selected to explore landslide deformation.

8. Methodology P5_26, It is not strict to conclude GA-LSSVM model has higher accuracy than other models due to the consideration of the trigger factors. Some other models also consider the trigger factors.

Response: Thank you for the careful reading and constructive comment. We have revised the text accordingly. Please see Methodology: Pg 5_line 27-28 in this revision.

9. Methodology P6, Fig 2, The technical route of left part is not clear. The methodology section is too long, authors are suggested to focus the introduction on what is new and what is developed by the authors to use the methodology to predict landslide displacement.

Response: Thank you for the careful reading and constructive comment. We have revised the text accordingly. Please see Methodology: Pg 6_Fig. 6 in this revision. In addition, Methodology P4_22-P5_2 of original manuscript, these are mentioned in Introduction: Pg 2_line 24-31 in this revision.

Case study P6_7,P6_17,P6_20-21,P6_25,P6_27-28, language should be improved.

Response: Thank you for the careful reading and constructive comments. We have revised the text accordingly. Please see Case study P6_8, P6_17-20, P6_26 in this revision.

11. Case study P7, Fig.4,& P8, Fig.5,the numbers in the legend needs to be explained.

Response: Thank you for the careful reading and constructive comments. About Fig. 4 and Fig.5, we have added the key for legend information accordingly. Please see Case Study: Pg 7_Fig.4 & Pg8_Fig.5 in this revision for details.

12. Case study P8_9-10, "in frontal area were relatively low" and "in the middle-rear areas were very high" are not consist with the monitoring data.

Response: Thank you for the careful reading. That is really a mistake in writing. We are very sorry for that our mistake. We have revised the text accordingly. Please see Case Study: Pg 8_line 12 and line 14 in this revision for details.

13. Case study P9_5, The location of the local road in fig.7 is suggested to be marked on the map.

Response: Thank you for the careful reading and constructive comment. We have revised the text accordingly. Please see Case study: Pg 10_Fig. 7 in this revision.

14. Case study P9_17-21, There is no groundwater monitoring method or data mention Here.

Response: Thank you for the careful reading and constructive comment. We have revised the text accordingly. Please see Case study: Pg 9_line 17-18 and Fig. 6 in this revision.

15. Landslide displacement prediction P11_5-6, "The model. . .regarding. . .", language should be improved.

Response: Thank you for the careful reading and constructive comment. That is really a spelling mistake. We are very sorry for that our mistake in spelling words. We have revised the text accordingly. Please see Landslide displacement prediction: Pg 11_line 5-6 in this revision.

16. Landslide displacement prediction P11_17-19, R2 are calculated according to the total data or to the predictive part of the data? Fig.9 is suggested to mark the R2, calculated according to the predictive part of the data, on the curves.

Response: Thank you for the careful reading and constructive comments. We agreed with the reviewer about the suggestion. We have revised the text accordingly. Please see Landslide displacement prediction: Pg 11_line 16-21 in this revision.

17. Landslide displacement prediction P12_11-16, " slight lag" is not described clearly.

Response: Thank you for the careful reading. In this submission, we have revised the text accordingly. Please see Landslide displacement prediction: Pg 12_line 15-19 and Pg 12_line 22-24 in this revision. Specific responses to the review comments are listed below.

From April 2007, it began to deform gently, giving rise (at station ZG86) to a maximum displacement of 184 mm over the 4-month period in May, June, July and August. Then from September, the periodic displacement of the landslide started to fall. During February and June 2007, the reservoir level decreased 10 m, while the rainfall was 297.7 mm during the subsequent 2 months, which should have been enough to trigger landslide deformation. Hence, the decrease of the reservoir water level continued to have an effect on displacement and there was also a lag effect, which means the displacement did not occur as soon as the reservoir water level decreased, but was delayed. As time passed, the effect of rainfall on the displacement diminished. From June to September 2007, the reservoir level remained stable, but displacement was still 115 mm, which demonstrated the lag effect of the influence of reservoir water level. When the reservoir water level and the groundwater depth are decreasing at different speeds, the groundwater will respond with a lag in relation to the variations of the reservoir water level. Because of this, falls between the levels increase in magnitude, which will make the hydrodynamic pressure in the landslide grow continuously and induce the most significant deformation of the year. This means that every decline in reservoir water level results in a rise in the curve of cumulative displacement of the landslide. Seen in terms of the spatial evolution of the landslide deformation, the surface cracks of the Shuping landslide develop during April and August every year and have mainly been located on the road at the middle-frontal areas of the landslide. There is also substantial consistency between cumulative displacement and the development of cracks: the greater the deformation, the more seriously the cracks developed.

18. Landslide displacement prediction P12 Fig.10, P13 Fig.11, why the authors choose the current month and past two month as two time periods for the indexes of variation of reservoir water level and rainfall? Is this choice reasonable? Because the influence period should be determined by detailed analyzing on the respond relationship between landslide displacement and influence factors.

Response: Thank you for the constructive comment. In this submission, we have revised the text accordingly. Please see Landslide displacement prediction: Pg 14_ line 2-3 and line 6-8 in this revision. Specific responses to the review comments are listed below.

In the Three Gorges Reservoir area, the external factors, including the reservoir water level, the rainfall and the groundwater, are the most significant transient forces that act upon landslides (Chen et al. 2005). In addition, as the identification of the landslide stability states may also be approached through the history of slope movements (Crozier 1986), the prophase displacement of landslides is also an essential item in the prediction of movement. Based on research on the relationship between landslide and reservoir water level (Please see Landslide displacement prediction-The predicted periodic component displacement: Pg 11_line 25 and Pg 12_line 1-19 in this revision), the variation of the reservoir water level 1 and 2 months before failure has a strong influence on landslide deformation rates. As shown in Fig. 10, over the previous 1 and 2 months, there are close relationships between variation of reservoir water level and the velocity of displacement. Change of reservoir level during the last month to reflect the influence of the rapidity of reservoir water level regulation: the analysis assumed that the water level was increased/decreased at constant velocity during 1 month. Based on research on the relationship between landslide and rainfall (Please see Landslide displacement prediction-The predicted periodic component displacement: Pg 11_line 25 and Pg 12_line 1-19 in this revision), the rainfall 1 and 2 months before failure has a strong influence on landslide deformation rates (Keefer et al. 1987; Zhang 2006; Du et al. 2013; Cao et al. 2015). As shown in Fig. 11, over the previous 1 and 2 months, there are also close relationships between cumulative rainfall and the velocity of displacement. Therefore, by detailed analyzing on the respond relationship between landslide displacement and influence factors, we choose the current month and past two month as two time periods for the indexes of variation of reservoir water level and rainfall.

References: Chen, J., Yang, Z.F., Li, X.: (2005) Relationship between landslide probability and rainfall in Three Gorges Reservoir area, Chin J Rock Mech Eng, 17, 7, 2005. Crozier, M.J.: Landslides: causes, consequences and environment, Croom Helm, London, 1986. Keefer, D.K., Wilson, R.C., Mark, R.K., Brabb, E.E., Brown, W.M., Ellen, S.D., Harp, E.L., Wieczorek, G.F., Alger, C.S., Zatkin, R.S.: Real-time landslide warning during heavy rainfall, Science, 238, 921–925, 1987. Zhang, G.R.: Spatial prediction and real-time warning of landslides and it's risk management based on WEBGIS, China University of Geosciences, Wuhan, 2006. Cao, Y., Yin, K., Alexander, D. E., and Zhou, C.: Using an extreme learning machine to predict the displacement of step-like landslides in relation to controlling factors, Landslides, 4, 725-736, 10.1007/s10346-015-0596-z, 2016. Du, J., Yin, K., and Lacasse, S.: Displacement prediction in colluvial landslides, Three Gorges Reservoir, China, Landslides, 10, 203-218, 10.1007/s10346-012-0326-8, 2013.

19. Landslide displacement prediction P13_16-18," the cumulative rainfall in the current month, the cumulative rainfall in the past two months, the reservoir water level, the variation in the reservoir water level in the current month, the variation in the reservoir water level in the past two months, and groundwater depth are selected as input variables", these variables have strong correlation, for instance, the reservoir water level and groundwater depth. Will this kind of dependent relationship between the variables influence the accuracy of prediction? How the authors think about it?

Response: Thank you for the constructive comment. We have revised the text accordingly. Please see Landslide displacement prediction: Pg 14_line 11-18 in this revision. Specific responses to the review comments are listed below.

Based on the analysis of monitoring data and deformation characteristics, rainfall and reservoir water level are the main factors that influence the landslide displacement. The reason is that the infiltration of rainfall and reservoir water level changes the dynamic characteristics of groundwater in landslide, which reflects the change of groundwater level. On the one hand, the change of groundwater level makes sliding mass or sliding zone in a dry and wet circulation state, which leads to changes in the physical and mechanical properties of the sliding mass or sliding zone. On the other hand, due to the change of groundwater level, the seepage force and the uplift pressure of groundwater acting on the landslide change dynamically. Due to differences in permeability of different landslides or the different parts of the landslide, the responses of groundwater level to rainfall and reservoir water level are not the same. In addition, the change in groundwater level exhibits considerable agreement with rainfall and reservoir water level fluctuations, with a slight lag. The relationships between the periodic displacement and the groundwater and the reservoir water level are illustrated in Fig. 10 and 11. For instance, in the rising phase of reservoir water level, the groundwater level gradually increases, with a slight lag behind the increase in the reservoir water level. The groundwater remains high enough for ongoing movement to continue. Conversely, the groundwater level decreases in the declining phase of the reservoir water level.

Therefore, groundwater influences displacement. Hence, considering the influences of rainfall and reservoir water level on landslide displacement, and in order to make prediction performance more accurate, it is necessary to select groundwater level as input variable for landslide prediction.

20. Landslide displacement prediction P14_9-10,"Notably,...water level." However, Fig.12 (b) did not match well.

Response: Thank you for the careful reading and constructive comment. We have revised the text accordingly. Please see Landslide displacement prediction: Pg 14_line 32-35 in this revision.

21. Landslide displacement prediction P14-16 Table 3, Table 4, Table 5, the measured cumulative displacement data are not from small to large in time. For example, ZG85ïïjN2012/11/1, 3442.907mm is smaller than 2012/10/1, 3460.208mm. Please explain why the cumulative displacement decreased?

Response: Thank you for the careful reading and constructive comment. To avoid the potential confusion, essential discussion was added in this revision.

It was noted that there are three situations like this. For example, in Table 3, 2012/11/1, 3442.907 mm is smaller than 2012/10/1, 3460.208 mm, and 2013/3/1, 3460.208 mm is smaller than 2013/2/1, 3477.509 mm. In Table 4, 2013/9/1, 4602.076 mm and 2013/10/1, 4602.076 mm are all smaller than 2013/8/1, 4619.377 mm. However, in Table 5, the measured cumulative displacement data are from small to large in time. To address the comments by the reviewer, the following explanations are made in this submission. Due to the rapid deformation rate of Shuping landslide in the present, Shuping landslide may deform and even fail. Hence, in order to prevent the deformation and failure of the landslide during the recession period of reservoir water level or heavy rainfall, it is necessary to take measurements to reduce the landslide displacement rate and improve the stability of the landslide. It is noted that from August 2012 to December 2012, Hubei Province geological environment Terminus had completed the investigation and emergency control design of Shuping landslide. The overall plan for emergency control engineering of Shuping landslide is to cut slope in middle-rear areas, press foot in frontal areas below 175 m of the reservoir water level and layout the drain ditch in the surface near the landslide boundary. After starting the emergency control project in 2013, by comparing the deformation characteristics of the landslide before and after the implementation of the project, the displacement rate decreased and the abrupt change of landslide disappeared gradually during the recession period of reservoir water level, which indicated that the emergency control project has achieved good performances. In addition, in the rising phase of the reservoir water level, the anti-sliding sections of the landslide are constantly submerged, and the anti-sliding force decreases while the sliding force decreases continuously at a faster rate. Moreover, due to the reverse seepage function of reservoir water level, it is beneficial to the stability of the landslide with characterizing of unobvious deformation of the landslide. Conversely, in the declining phase of the reservoir water level, the sliding force increases under the action of the hydrodynamic pressure in the anti-sliding sections of the landslide, which results in the large deformation of the landslide and the decline of the stability. In terms of this kind of landslide, there is basically no sign of deformation in the rising phase of the reservoir water level. But it is very sensitive to the decline of reservoir water level. For instance, in October 2012, the reservoir water level rose close to higher value (172.7 m) than that (166.5 m) in November 2012. In this month, the deformation velocity of the front edge (ZG85) decreased to 0.56 mm/day. Hence, for ZG85, the cumulative displacement on 2012/11/1 is smaller than that on 2012/10/1. During the subsequent water level decreasing stage in July 2013, the water level decreased to the lowest value (145 m). The deformation velocity of the front edge increased to 3.58 mm/day. In conclusion, the rising phase and the declining phase of the reservoir water level will make landslide displacement vibration. That means that the trend of displacement growth of some landslides is not obvious, with characterizing of jagged cumulative displacement curve related to the rising phase and the declining phase of the reservoir water level. Therefore, the measured cumulative displacement data are not necessarily from small to large in time. Such is the displacement of Sangshuping landslide, which is also located in Shazhenxi town, Zigui country, Hubei province, China, near the tributary of Yangtze River, Qinggan River and approximately 6 km into the Yangtze River. Moreover, according to the reference (Huang et al. 2016), the cumulative displacement of the Baijiabao landslide monitored by GPS are fluctuation within a narrow range and is not monotonically increasing with time, such as in August 2007 and June 2009.

References: Huang, F. M., Yin, K. L., Zhang G. R., Gui, L., Yang, B. B., Liu, L.: (2016) Landslide displacement prediction using discrete wavelet transform and extreme learning machine based on chaos theory, Environmental Earth Sciences, 75, 20, 1376.

Please also note the supplement to this comment:
https://www.nat-hazards-earth-syst-sci-discuss.net/nhess-2017-87/nhess-2017-87-AC3-supplement.pdf

**Supplement:**

**Response to Reviewer's Comments**

The authors would like to thank the reviewer for the careful reading and constructive comments that have helped sharpen this manuscript. In this revision, all the comments of the reviewers have been carefully addressed. Specific responses to the review comments are listed below. The line numbers refer to those in the revised manuscript.

**Anonymous Referee #2**

*1. Introduction P1_29, "…external factors, such as geological conditions…"ïïjN geological C1 conditions should be internal factors.*

Response: Thank you for the careful reading. Indeed, geological conditions are considered as an internal factor in the manuscript. We have revised the manuscript accordingly. Please see **Introduction: Pg 1_lines 35-36** in this revision.

*2. Introduction P1_36,"in recently years" should be " in recent years".*

Response: Thank you for the careful reading. We have revised the text accordingly. Please see **Introduction: Pg 1_lines 40** in this revision.

*3. Introduction P2_23-26, Here, the studies, which also used GA-LSSVM model to predict landslide displacement, are suggested to be mentioned. For example, Cai Z, Xu W, Meng Y, et al. Prediction of landslide displacement based on GA-LSSVM with multiple factors. Bulletin of Engineering Geology & the Environment, 2016, 75(2):637-646.*

Response: Thank you for the constructive comments. We have revised the text accordingly. Please see **Introduction: Pg 2_line 34** in this revision.

*4. Methodology P3_24, "By searching or a function…"ïïjNhere "or" I guess is a spelling mistake.*

Response: Thank you for the careful reading. That is really a spelling mistake. We are very sorry for that our mistake in spelling words. We have revised the text accordingly. Please see **Methodology: Pg 3_line 34** in this revision.

*5. Methodology P4_17-18, I suggest the authors to supplement an equation contains both C and $\sigma$ to express the model.*

Response: Thank you for the careful reading and constructive comments. We have revised the text accordingly. Please see **Methodology: Pg 5_line 1-2** in this revision.

It can be seen from this paper that $C$ is a penalty factor representing the penalty degree of the training samples, and $\sigma$ is a parameter of the kernel function. The parameter of the model $C$ and the parameter of the kernel function $\sigma$ significantly influence the prediction performance. The parameter

*C* represents the error tolerance. The more accurate the parameter is, the higher the prediction performance is, but this can lead to overtraining. The parameter $\sigma$ implicitly determines the spatial distribution of data mapping in the new feature space.

In this paper, the radial kernel function is selected as the kernel function in the LSSVM model, so the determination of the parameters C and σ is very significant for the great prediction performance. However, the equation between C and σ expressed jointly the Eq. (5) ~ (10) is extremely complicated, which is inconvenient to be expressed by a certain formula.

In the machine learning of LSSVM, the parameters C and σ are hyper parameters that set the values before the beginning of the learning process, rather than the parameter data obtained by training. In general, it is necessary to optimize the hyper parameters and select a set of optimal hyper parameters for the machine learning of LSSVM to improve the performance and effective of learning. In this paper, the GA is selected as the method of parameter optimization in the LSSVM due to its advantages in determining the unknown parameters that are consistent between the predicted data and the measured data. By introducing the GA, the parameters C and σ can be derived automatically. In the process of calculation, the best parameter C is first obtained by searching the optimum. Then based on the best parameter C, the best parameter σ is obtained by training.

*6. Methodology P4_22-P5_2, These could be mentioned in introduction or put forward in a discussion section.*

Response: Thank you for the careful reading and kind comment. We have revised the text accordingly. Please see **Introduction: Pg 2_line 24-31** in this revision.

*7. Methodology P5_23-24, "The sampling…sampling data." This sentence is confusing. Why the data is independent.*

Response: Thank you for the careful reading and constructive comment. We have revised the text accordingly. Indeed, the sampling data used for landslide displacement prediction are continuous and mutually dependent landslide data which are applicable or feasible to the specific method. We are very sorry for that our mistake in this sentence. Please see **Introduction: Pg 3_line 1-6 and Methodology: Pg 5_line 25** in this revision.

It is well known that the evolution process of landslide is a complex non-linear process that is caused by the complex interaction of different factors, e.g. the complicated geological settings, varying hydrological conditions. Displacement time series are generally appreciated as the direct representation of the complex and non-linear dynamical behaviour of landslide. However, the landslide displacement induced by the exteral factors is approximately periodic. Therefore, a landslide displacement sequence is an instability time series with a periodic episodic movement characteristic. Because the integrity of the data collected at monitoring points has an effect on the displacement prediction, the monitoring data from July 2003 to October 2013 are selected to explore landslide deformation.

*8. Methodology P5_26, It is not strict to conclude GA-LSSVM model has higher accuracy than other models due to the consideration of the trigger factors. Some other models also consider the trigger factors.*

Response: Thank you for the careful reading and constructive comment. We have revised the text accordingly. Please see **Methodology: Pg 5_line 27-28** in this revision.

*9. Methodology P6, Fig 2, The technical route of left part is not clear. The methodology section is too long, authors are suggested to focus the introduction on what is new and what is developed by the authors to use the methodology to predict landslide displacement.*

Response: Thank you for the careful reading and constructive comment. We have revised the text accordingly. Please see **Methodology: Pg 6_Fig. 6** in this revision. In addition, **Methodology P4_22-P5_2** of original manuscript, these are mentioned in **Introduction: Pg 2_line 24-31** in this revision.

*10 Case study P6_7,P6_17,P6_20-21,P6_25,P6_27-28, language should be improved.*

Response: Thank you for the careful reading and constructive comments. We have revised the text accordingly. Please see **Case study P6_8, P6_17-20, P6_26** in this revision.

*11. Case study P7, Fig.4,& P8, Fig.5,the numbers in the legend needs to be explained.*

Response: Thank you for the careful reading and constructive comments. About Fig. 4 and Fig.5, we have added the key for legend information accordingly. Please see **Case Study: Pg 7_Fig.4 & Pg8_Fig.5** in this revision for details.

*12. Case study P8_9-10, "in frontal area were relatively low" and "in the middle-rear areas were very high" are not consist with the monitoring data.*

Response: Thank you for the careful reading. That is really a mistake in writing. We are very sorry for that our mistake. We have revised the text accordingly. Please see **Case Study: Pg 8_line 12 and line 14** in this revision for details.

*13. Case study P9_5, The location of the local road in fig.7 is suggested to be marked on the map.*

Response: Thank you for the careful reading and constructive comment. We have revised the text accordingly. Please see **Case study: Pg 10_Fig. 7** in this revision.

*14. Case study P9_17-21, There is no groundwater monitoring method or data mention Here.*

Response: Thank you for the careful reading and constructive comment. We have revised the text accordingly. Please see **Case study: Pg 9_line 17-18 and Fig. 6** in this revision.

*15. Landslide displacement prediction P11_5-6, "The model…regarding…", language should be improved.*

Response: Thank you for the careful reading and constructive comment. That is really a spelling mistake. We are very sorry for that our mistake in spelling words. We have revised the text accordingly. Please see **Landslide displacement prediction: Pg 11_line 5-6** in this revision.

**16. Landslide displacement prediction P11_17-19, R2 are calculated according to the total data or to the predictive part of the data? Fig.9 is suggested to mark the R2, calculated according to the predictive part of the data, on the curves.**

Response: Thank you for the careful reading and constructive comments. We agreed with the reviewer about the suggestion. We have revised the text accordingly. Please see **Landslide displacement prediction: Pg 11_line 16-21** in this revision.

**17. Landslide displacement prediction P12_11-16, " slight lag" is not described clearly.**

Response: Thank you for the careful reading. In this submission, we have revised the text accordingly. Please see **Landslide displacement prediction: Pg 12_line 15-19 and Pg 12_line 22-24** in this revision. Specific responses to the review comments are listed below.

From April 2007, it began to deform gently, giving rise (at station ZG86) to a maximum displacement of 184 mm over the 4-month period in May, June, July and August. Then from September, the periodic displacement of the landslide started to fall. During February and June 2007, the reservoir level decreased 10 m, while the rainfall was 297.7 mm during the subsequent 2 months, which should have been enough to trigger landslide deformation. Hence, the decrease of the reservoir water level continued to have an effect on displacement and there was also a lag effect, which means the displacement did not occur as soon as the reservoir water level decreased, but was delayed. As time passed, the effect of rainfall on the displacement diminished.

From June to September 2007, the reservoir level remained stable, but displacement was still 115 mm, which demonstrated the lag effect of the influence of reservoir water level. When the reservoir water level and the groundwater depth are decreasing at different speeds, the groundwater will respond with a lag in relation to the variations of the reservoir water level. Because of this, falls between the levels increase in magnitude, which will make the hydrodynamic pressure in the landslide grow continuously and induce the most significant deformation of the year. This means that every decline in reservoir water level results in a rise in the curve of cumulative displacement of the landslide. Seen in terms of the spatial evolution of the landslide deformation, the surface cracks of the Shuping landslide develop during April and August every year and have mainly been located on the road at the middle-frontal areas of the landslide. There is also substantial consistency between cumulative displacement and the development of cracks: the greater the deformation, the more seriously the cracks developed.

**18. Landslide displacement prediction P12 Fig.10, P13 Fig.11, why the authors choose the current month and past two month as two time periods for the indexes of variation of reservoir water level and rainfall? Is this choice reasonable? Because the influence period should be determined by detailed analyzing on the respond relationship between landslide displacement and influence factors.**

Response: Thank you for the constructive comment. In this submission, we have revised the text accordingly. Please see **Landslide displacement prediction: Pg 14_ line 2-3 and line 6-8** in this revision. Specific responses to the review comments are listed below.

In the Three Gorges Reservoir area, the external factors, including the reservoir water level, the rainfall and the groundwater, are the most significant transient forces that act upon landslides (Chen et al. 2005). In addition, as the identification of the landslide stability states may also be approached through the history of slope movements (Crozier 1986), the prophase displacement of landslides is also an essential item in the prediction of movement.

Based on research on the relationship between landslide and reservoir water level (Please see **Landslide displacement prediction-The predicted periodic component displacement: Pg 11_line 25 and Pg 12_line 1-19** in this revision), the variation of the reservoir water level 1 and 2 months before failure has a strong influence on landslide deformation rates. As shown in Fig. 10, over the previous 1 and 2 months, there are close relationships between variation of reservoir water level and the velocity of displacement. Change of reservoir level during the last month to reflect the influence of the rapidity of reservoir water level regulation: the analysis assumed that the water level was increased/decreased at constant velocity during 1 month.

Based on research on the relationship between landslide and rainfall (Please see **Landslide displacement prediction-The predicted periodic component displacement: Pg 11_line 25 and Pg 12_line 1-19** in this revision), the rainfall 1 and 2 months before failure has a strong influence on landslide deformation rates (Keefer et al. 1987; Zhang 2006; Du et al. 2013; Cao et al. 2015). As shown in Fig. 11, over the previous 1 and 2 months, there are also close relationships between cumulative rainfall and the velocity of displacement.

Therefore, by detailed analyzing on the respond relationship between landslide displacement and influence factors, we choose the current month and past two month as two time periods for the indexes of variation of reservoir water level and rainfall.

Based on the analysis of monitoring data and deformation characteristics, rainfall and reservoir water level are the main factors that influence the landslide displacement. The reason is that the infiltration of rainfall and reservoir water level changes the dynamic characteristics of groundwater in landslide, which reflects the change of groundwater level. On the one hand, the change of groundwater level makes sliding mass or sliding zone in a dry and wet circulation state, which leads to changes in the physical and mechanical properties of the sliding mass or sliding zone. On the other hand, due to the change of groundwater level, the seepage force and the uplift pressure of groundwater acting on the landslide change dynamically.

Due to differences in permeability of different landslides or the different parts of the landslide, the responses of groundwater level to rainfall and reservoir water level are not the same. In addition, the change in groundwater level exhibits considerable agreement with rainfall and reservoir water level fluctuations, with a slight lag. The relationships between the periodic displacement and the groundwater and the reservoir water level are illustrated in Fig. 10 and 11. For instance, in the rising phase of reservoir water level, the groundwater level gradually increases, with a slight lag behind the increase in the reservoir water level. The groundwater remains high enough for ongoing movement to continue. Conversely, the groundwater level decreases in the declining phase of the reservoir water level. Therefore, groundwater influences displacement.

Hence, considering the influences of rainfall and reservoir water level on landslide displacement, and in order to make prediction performance more accurate, it is necessary to select groundwater level as input variable for landslide prediction.

*20. Landslide displacement prediction P14_9-10,"Notably,…water level." However, Fig.12 (b) did not match well.*

Response: Thank you for the careful reading and constructive comment. We have revised the text accordingly. Please see **Landslide displacement prediction: Pg 14_line 32-35** in this revision.

*21. Landslide displacement prediction P14-16 Table 3, Table 4, Table 5, the measured cumulative displacement data are not from small to large in time. For example, ZG85üjN2012/11/1, 3442.907mm is smaller than 2012/10/1, 3460.208mm. Please explain why the cumulative displacement decreased?*

Response: Thank you for the careful reading and constructive comment. To avoid the potential confusion, essential discussion was added in this revision.

It was noted that there are three situations like this. For example, in Table 3, 2012/11/1, 3442.907 mm is smaller than 2012/10/1, 3460.208 mm, and 2013/3/1, 3460.208 mm is smaller than 2013/2/1, 3477.509 mm. In Table 4, 2013/9/1, 4602.076 mm and 2013/10/1, 4602.076 mm are all smaller than 2013/8/1, 4619.377 mm. However, in Table 5, the measured cumulative displacement data are from small to large in time. To address the comments by the reviewer, the following explanations are made in this submission.

Due to the rapid deformation rate of Shuping landslide in the present, Shuping landslide may deform and even fail. Hence, in order to prevent the deformation and failure of the landslide during the recession period of reservoir water level or heavy rainfall, it is necessary to take measurements to reduce the landslide displacement rate and improve the stability of the landslide. It is noted that from August 2012 to December 2012, Hubei Province geological environment Terminus had completed the investigation and emergency control design of Shuping landslide. The overall plan for emergency control engineering of Shuping landslide is to cut slope in middle-rear areas, press foot in frontal areas below 175 m of the reservoir water level and layout the drain ditch in the surface near the landslide boundary. After starting the emergency control project in 2013, by comparing the deformation characteristics of the landslide before and after the implementation of the project, the displacement rate decreased and the abrupt change of landslide disappeared gradually during the recession period of reservoir water level, which indicated that the emergency control project has achieved good performances.

In addition, in the rising phase of the reservoir water level, the anti-sliding sections of the landslide are constantly submerged, and the anti-sliding force decreases while the sliding force decreases continuously at a faster rate. Moreover, due to the reverse seepage function of reservoir water level, it is beneficial to the stability of the landslide with characterizing of unobvious deformation of the landslide. Conversely, in the declining phase of the reservoir water level, the sliding force increases under the action of the hydrodynamic pressure in the anti-sliding sections of the landslide, which results in the large deformation of the landslide and the decline of the stability. In terms of this kind of landslide, there is basically no sign of deformation in the rising phase of the reservoir water level. But it is very sensitive to the decline of reservoir water level.

For instance, in October 2012, the reservoir water level rose close to higher value (172.7 m) than that (166.5 m) in November 2012. In this month, the deformation velocity of the front edge (ZG85) decreased to 0.56 mm/day. Hence, for ZG85, the cumulative displacement on 2012/11/1 is smaller than that on 2012/10/1. During the subsequent water level decreasing stage in July 2013, the water level decreased to the lowest value (145 m). The deformation velocity of the front edge increased to 3.58 mm/day.

In conclusion, the rising phase and the declining phase of the reservoir water level will make landslide displacement vibration. That means that the trend of displacement growth of some landslides is not obvious, with characterizing of jagged cumulative displacement curve related to the rising phase and the declining phase of the reservoir water level. Therefore, the measured cumulative displacement data are not necessarily from small to large in time. Such is the displacement of Sangshuping landslide, which is also located in Shazhenxi town, Zigui country, Hubei province, China, near the tributary of Yangtze River, Qinggan River and approximately 6 km into the Yangtze River. Moreover, according to the reference (Huang et al. 2016), the cumulative displacement of the Baijiabao landslide monitored by GPS are fluctuation within a narrow range and is not monotonically increasing with time, such as in August 2007 and June 2009.

References:

Huang, F. M., Yin, K. L., Zhang G. R., Gui, L., Yang, B. B., Liu, L.: (2016) Landslide displacement prediction using discrete wavelet transform and extreme learning machine based on chaos theory, Environmental Earth Sciences, 75, 20, 1376.

[revised manuscript text omitted]